# TASK-FREE ADAPTIVE META BLACK-BOX OPTIMIZATION

**Chao Wang**[1], **Licheng Jiao**[1], **Lingling Li**[1,*], **Jiaxuan Zhao**[2], **Guanchun Wang**[1], **Fang Liu**[1]
**Shuyuan Yang**[1]
[1]Xidian University   [2]Tianjin Research Institute for Water Transport Engineering, M.O.T.
`xiaofengxd@126.com, lchjiao@mail.xidian.edu.cn,`
`{llli,wangguanchun,syyang}@xidian.edu.cn,`
`jiaxuanzhao@stu.xidian.edu.cn, f63liu@163.com`

## ABSTRACT

Handcrafted optimizers become prohibitively inefficient for complex black-box optimization (BBO) tasks. MetaBBO addresses this challenge by meta-learning to automatically configure optimizers for low-level BBO tasks, thereby eliminating heuristic dependencies. However, existing methods typically require extensive handcrafted training tasks to learn meta-strategies that generalize to target tasks, which poses a critical limitation for realistic applications with unknown task distributions. To overcome the issue, we propose the Adaptive meta Black-box Optimization Model (ABOM), which performs online parameter adaptation using solely optimization data from the target task, obviating the need for predefined task distributions. Unlike conventional metaBBO frameworks that decouple meta-training and optimization phases, ABOM introduces a closed-loop adaptive parameter learning mechanism, where parameterized evolutionary operators continuously self-update by leveraging generated populations during optimization. This paradigm shift enables zero-shot optimization: ABOM achieves competitive performance on synthetic BBO benchmarks and realistic unmanned aerial vehicle path planning problems without any handcrafted training tasks. Visualization studies reveal that parameterized evolutionary operators exhibit statistically significant search patterns, including natural selection and genetic recombination.

## 1 INTRODUCTION

Black-box optimization (BBO) problems arise in diverse machine learning applications such as neuroevolution Stanley et al. (2019); Miikkulainen (2025), hyperparameter tuning Bai & Cheng (2024), neural architecture search Wang et al. (2023); Salmani Pour Avval et al. (2025), and prompt engineering Romera-Paredes et al. (2024); Wang et al. (2025a). In these scenarios, the objective function is accessible solely through expensive evaluations $f(x)$, with derivative information like gradients or Hessians inherently unavailable. Evolutionary algorithms (EAs) Eiben & Smith (2015); De Jong (2017) address this challenge by iteratively updating populations through derivative-free heuristic operators, including selection, crossover, and mutation, to explore complex fitness landscapes. Recent advances in computational infrastructure have enabled EAs to generate robust solutions for increasingly complex BBO problems Miikkulainen & Forrest (2021).

The "No Free Lunch" (NFL) theorem Wolpert & Macready (2002) establishes that no optimization algorithm universally outperforms others across all problem domains. To enhance cross-domain applicability, numerous adaptive mechanisms have been designed Bäck & Schwefel (1993); Brest et al. (2021); Li et al. (2013); Hansen (2016); Tao et al. (2021) that leverage optimization data generated during the search process to dynamically select operators or adjust parameters. Although these adaptive methods achieve strong performance on standard benchmarks, they require specialized expertise in optimization theory and problem characteristics Ma et al. (2024). Meta Black-Box Optimization (MetaBBO) addresses this limitation by automating meta-level strategies Ma et al. (2025b), such as algorithm selection Tian et al. (2020); Guo et al. (2024), algorithm configuration Lange et al. (2023b;a); Guo et al. (2025), solution manipulation Li et al. (2024; 2025), and generative design Chen et al. (2024); Yang et al. (2024), through meta-learning (Fig. 1, Left). Yet existing MetaBBO

methods require training on handcrafted task distributions $\mathcal{F}$ or prior knowledge for generalization to new domains. Since such distributions are often inaccessible in practical scenarios (e.g., when the target task is unique or data-scarce), this dependency severely limits real-world deployment.

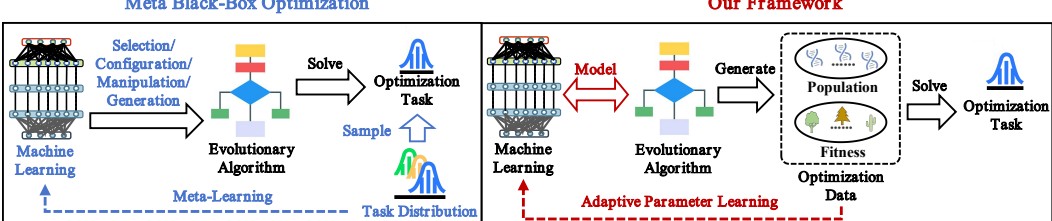

Figure 1: Conceptual comparison: (Left) MetaBBO methods learn meta-strategies from task distributions but depend on handcrafted training tasks; (Right) Our framework performs adaptive parameter learning using self-generated optimization data, eliminating task distribution dependency.

To address this limitation, we propose the Adaptive meta Black-box Optimization Model (ABOM), a task-free meta-optimizer that adaptively learns parameters using only self-generated data (Fig. 1, Right). ABOM's distinguishing feature is an end-to-end differentiable framework that parameterizes evolutionary operators as learnable functions (Fig. 2). Inspired by EA dynamics, it employs attention mechanisms to separately model relationships among individuals, fitness landscapes, and genetic components, thereby replicating selection, crossover, and mutation as differentiable operations. Crucially, ABOM updates its parameters during optimization by aligning the generated offspring population with an elite archive of high-quality solutions, bypassing the need for meta-training on task distributions. This design yields two key contributions:

- **Task-free adaptation**: The parameters of ABOM are updated via adaptive learning using optimization data from the target task, eliminating the reliance on handcrafted training tasks or heuristic rules. Theoretically, ABOM guarantees convergence to the global optimum.

- **Intrinsic interpretability**: Attention matrices provide quantifiable insights into search patterns, such as selection bias toward high-fitness individuals and consistent genetic interaction patterns during mutation. Moreover, ABOM supports GPU acceleration out of the box, without requiring changes to standard EA infrastructure.

## 2 RELATED WORKS

**Evolutionary Algorithms**. EAs, such as genetic algorithms (GA) Holland (1962), evolution strategies (ES)Rechenberg (1984), particle swarm optimization (PSO) Kennedy & Eberhart (1995), and differential evolution (DE) Storn & Price (1997), are widely adopted for BBO tasks due to their derivative-free nature. These methods manipulate populations via heuristic operators but often suffer from inefficiency and fragility when applied to new tasks, as they require labor-intensive manual parameter tuning. While ABOM draws inspiration from EA dynamics, it eliminates manual tuning by enabling adaptive parameter learning directly from optimization data.

**Adaptive Optimization**. To improve cross-domain generalization, adaptive EA variants employ dynamic operator selection or parameter adjustment such as CMAES Hansen (2016); Ollivier et al. (2017), SAHLPSO Tao et al. (2021), JDE21 Brest et al. (2021)). These methods achieve state-of-the-art results on standard BBO benchmarks but demand deep expertise in optimization theory and often require problem-specific GPU acceleration for scalability. In contrast, ABOM adheres to a unified deep learning architecture, replacing heuristic rules with adaptive parameter learning and reducing deployment barriers.

**Meta Black-Box Optimization**. MetaBBO techniques leverage meta-learning to automate meta-level strategies for solving lower-level BBO tasks Ma et al. (2025b); Wang et al. (2025b); Yun et al. (2025), thereby reducing the need for expert intervention. Common paradigms include algorithm selection Tian et al. (2020); Guo et al. (2024), which chooses from a predefined pool of operators; algorithm configuration Lange et al. (2023b;a); Guo et al. (2025), which tunes hyperparameters via

meta-strategies; solution manipulation Li et al. (2024; 2025), which integrates meta-strategies directly into the optimization process; and algorithm generation Chen et al. (2024); Yang et al. (2024), which synthesizes entire optimization workflows. Despite their promise, these methods critically depend on manually designed components, such as discrete algorithm search spaces $\mathcal{A}$, state feature spaces, meta-objectives, and training task distributions $\mathcal{F}$. The dependency on handcrafted $\mathcal{F}$ hinders real-world applicability when task distributions are unavailable. ABOM addresses this limitation by unifying evolutionary operators into a continuous, differentiable parameter space, enabling adaptive parameter learning without requiring $\mathcal{F}$ or discrete algorithm search spaces.

# 3 ADAPTIVE META BLACK-BOX OPTIMIZATION MODEL

## 3.1 PROBLEM DEFINITION

A target BBO task is defined as:

$$\min_{\mathbf{x} \in \mathbb{R}^d} f_T(\mathbf{x}), \tag{1}$$

where $\mathbf{x}$ is the solution vector in a $d$-dimensional search space. MetaBBO methods formalize the automated design of optimizers as a triplet $\mathcal{T} := (\mathcal{A}, \mathcal{R}, \mathcal{F})$, with the discrete algorithm search space $\mathcal{A}$, the performance metric $\mathcal{R}$, and the training task distribution $\mathcal{F}$. The meta-optimization objective maximizes expected performance Ma et al. (2025b):

$$J(\boldsymbol{\theta}) = \max_{\boldsymbol{\theta} \in \boldsymbol{\Theta}} \mathbb{E}_{f \sim \mathcal{F}} \left[ \mathcal{R}(\mathcal{A}, \pi_{\boldsymbol{\theta}}, f) \right], \tag{2}$$

where the meta-strategy $\pi_{\boldsymbol{\theta}}$ selects the algorithm (or configuration) $a \in \mathcal{A}$ for each task $f$. The Eq. (2) needs to be designed manually $\mathcal{F}$. To mitigate the need for $\mathcal{F}$, we define adaptive MetaBBO as $\mathcal{T}_{\text{adaptive}} := (\mathcal{A}, \mathcal{R}, f_T)$, operating directly on the target task $f_T$. Using cumulative optimization knowledge $\mathcal{M}^{(t)} = (\mathcal{X}^{(t)}, \mathcal{Y}^{(t)})$, where $\mathcal{X}^{(t)} = \{\mathbf{x}^{(1)}, \ldots, \mathbf{x}^{(t)}\}$ (solutions) and $\mathcal{Y}^{(t)} = \{f(\mathbf{x}^{(1)}), \ldots, f(\mathbf{x}^{(t)})\}$ (evaluations) during optimization, the Eq. (2) becomes the following:

$$J(\boldsymbol{\theta}) = \max_{\boldsymbol{\theta} \in \boldsymbol{\Theta}} \left[ \mathcal{R}(\mathcal{A}, \pi_{\boldsymbol{\theta}}, \mathcal{M}^{(t)}) \right], \tag{3}$$

with $\boldsymbol{\theta}$ updated online using $\mathcal{M}^{(t)}$. However, $\mathcal{A}$ and $\pi_{\boldsymbol{\theta}}$ still require expert-crafted components.

To address this limitation, ABOM replaces the discrete meta-optimization framework $(\mathcal{A}, \pi_{\boldsymbol{\theta}})$ with a single, differentiable optimizer $\pi_{\boldsymbol{\theta}}$ parameterized by $\boldsymbol{\theta}$. The final objective is:

$$J(\boldsymbol{\theta}) = \max_{\boldsymbol{\theta} \in \boldsymbol{\Theta}} \left[ \mathcal{R}(\pi_{\boldsymbol{\theta}}, \mathcal{M}^{(t)}) \right], \tag{4}$$

where $\boldsymbol{\theta}$ is updated *only* using $\mathcal{M}^{(t)}$ from $f_T$, thereby eliminating the need for manual design of $\mathcal{F}$, discrete search spaces $\mathcal{A}$, and expert-dependent feature engineering. The Eq. (4) establishes an end-to-end differentiable framework where adaptive parameter learning occurs through continuous feedback from $\mathcal{M}^{(t)}$.

## 3.2 META-STRATEGY ARCHITECTURE

ABOM implements a differentiable meta-strategy $\hat{\mathbf{P}}^{(t)} = \pi_{\boldsymbol{\theta}}(\mathbf{P}^{(t)}, \mathbf{F}^{(t)})$ (Fig. 2, Bottom) that learns evolutionary operators via attention mechanisms Vaswani et al. (2017). At generation $t$, the population $\mathbf{P}^{(t)} = \left[ \mathbf{p}_1^{(t)\top}; \ldots; \mathbf{p}_N^{(t)\top} \right] \in \mathbb{R}^{N \times d}$ represents $N$ candidate solutions in the search space, where the individual $\mathbf{p}_i^{(t)} \in \mathbb{R}^d$ is a solution vector. The fitness values $\mathbf{F}^{(t)} = \left[ f_T(\mathbf{p}_1^{(t)}); \ldots; f_T(\mathbf{p}_N^{(t)}) \right] \in \mathbb{R}^N$ are scalar evaluations $f_T(\mathbf{p}_i^{(t)})$ obtained via black-box queries to the target objective $f_T(\cdot)$, with lower values indicating better solutions. Given the population-fitness pair, ABOM generates offspring through three unified modules:

**Selection.** The selection matrix $\mathbf{A}^{(t)} \in \mathbb{R}^{N \times N}$ is computed to jointly model relationships in the solution space and among fitness values via attention:

$$\mathbf{A}^{(t)} = \text{softmax} \left( \frac{(\mathbf{P}^{(t)} \mathbf{W}^{QP})(\mathbf{P}^{(t)} \mathbf{W}^{KP})^\top + (\mathbf{F}^{(t)} \mathbf{W}^{QF})(\mathbf{F}^{(t)} \mathbf{W}^{KF})^\top}{\sqrt{d_A}} \right), \tag{5}$$

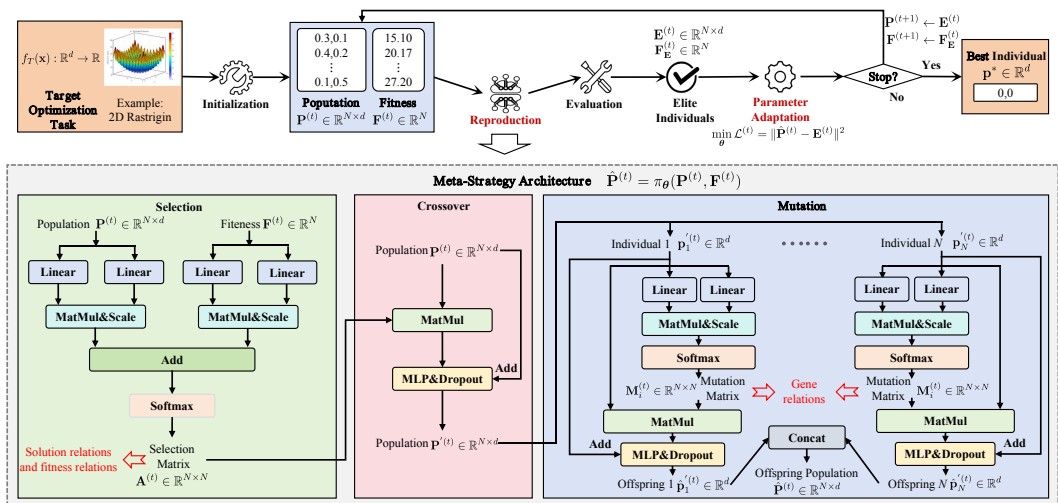

Figure 2: Workflow of ABOM: (Top) Adaptive optimization loop: Initialization, reproduction, evaluation, elitism, and parameter adaptation; (Bottom) Meta-strategies for reproduction: Attention-based evolutionary operators, including selection, crossover, and mutation.

where $\mathbf{W}^{QP}, \mathbf{W}^{KP} \in \mathbb{R}^{d \times d_A}$ project solution features, and $\mathbf{W}^{QF}, \mathbf{W}^{KF} \in \mathbb{R}^{1 \times d_A}$ process fitness values. The first term captures spatial relationships in the solution space, while the second term encodes fitness-driven selection pressure. This dual-path design ensures that recombination prioritizes solutions based on both their search-space positioning and fitness ranking, rather than fitness alone.

**Crossover.** The intermediate population $\mathbf{P}'^{(t)}$ is generated by:

$$\mathbf{P}'^{(t)} = \mathbf{P}^{(t)} + \mathrm{MLP}_{\theta_c}\left(\mathbf{A}^{(t)}\mathbf{P}^{(t)}\right), \tag{6}$$

where $\mathrm{MLP}_{\theta_c}(\mathbf{z}) = \tanh(\mathbf{z}\mathbf{W}_1 + \mathbf{b}_1)\mathbf{W}_2 + \mathbf{b}_2$ with $\mathbf{W}_1 \in \mathbb{R}^{d \times d_M}$, $\mathbf{b}_1 \in \mathbb{R}^{d_M}$, $\mathbf{W}_2 \in \mathbb{R}^{d_M \times d}$, $\mathbf{b}_2 \in \mathbb{R}^d$. Dropout with rate $p_C$ is applied to the hidden layer during both adaptive parameter learning and inference. The mechanism ensures persistent exploration through controlled randomness and is consistently maintained across all stochastic operations in ABOM. The term $\mathbf{A}^{(t)}\mathbf{P}^{(t)}$ computes an adaptive recombination pool: each row $\sum_{j=1}^{N} \mathbf{A}_{i,j}^{(t)}\mathbf{p}_j^{(t)}$ represents a context-aware blend of parent solutions, where weights $\mathbf{A}_{i,j}^{(t)}$ dynamically balance proximity in solution space and fitness-driven selection pressure.

**Mutation.** For each individual $\mathbf{p}_i'^{(t)} \in \mathbb{R}^d$ in $\mathbf{P}'^{(t)}$, offspring $\hat{\mathbf{p}}_i^{(t)}$ is generated via:

$$\hat{\mathbf{p}}_i^{(t)} = \mathbf{p}_i'^{(t)} + \mathrm{MLP}_{\theta_m}\left(\mathbf{M}_i^{(t)}\mathbf{p}_i'^{(t)}\right), \quad \mathbf{M}_i^{(t)} = \mathrm{softmax}\left(\frac{(\mathbf{p}_i'^{(t)}\mathbf{W}^{QM})(\mathbf{p}_i'^{(t)}\mathbf{W}^{KM})^\top}{\sqrt{d_A}}\right), \quad (7)$$

where $\mathbf{W}^{QM}, \mathbf{W}^{KM} \in \mathbb{R}^{1 \times d_A}$, and $\mathrm{MLP}_{\theta_m}(\mathbf{z}) = \tanh(\mathbf{z}\mathbf{W}_3 + \mathbf{b}_3)\mathbf{W}_4 + \mathbf{b}_4$ with $\mathbf{W}_3 \in \mathbb{R}^{1 \times d_M}$, $\mathbf{b}_3 \in \mathbb{R}^{d_M}$, $\mathbf{W}_4 \in \mathbb{R}^{d_M}$, $\mathbf{b}_4 \in \mathbb{R}$. Following the same exploration principle as crossover, dropout with rate $p_M$ is applied during inference to maintain persistent exploration. The mutation matrix $\mathbf{M}^{(t)} \in \mathbb{R}^{d \times d}$ dynamically models gene-wise dependencies: each entry $\mathbf{M}_{j,k}^{(t)}$ quantifies the interaction strength between the $j$-th and $k$-th dimensions, enabling context-aware perturbations. Finally, offspring are concatenated as:

$$\hat{\mathbf{P}}^{(t)} = \left[\hat{\mathbf{p}}_1^{(t)\top}; \dots; \hat{\mathbf{p}}_N^{(t)\top}\right] \in \mathbb{R}^{N \times d}. \tag{8}$$

The set $\boldsymbol{\theta}$ containing all parameters is:

$$\boldsymbol{\theta} = \left\{\mathbf{W}^{QP}, \mathbf{W}^{KP}, \mathbf{W}^{QF}, \mathbf{W}^{KF}, \mathbf{W}^{QM}, \mathbf{W}^{KM}\right\} \cup \theta_c \cup \theta_m, \tag{9}$$

with $\theta_c = \{\mathbf{W}_1, \mathbf{b}_1, \mathbf{W}_2, \mathbf{b}_2\}$, $\theta_m = \{\mathbf{W}_3, \mathbf{b}_3, \mathbf{W}_4, \mathbf{b}_4\}$. Note that $p_C$ and $p_M$ are hyperparameters that govern the intensity of exploration. All modules share attention dimension $d_A$ and MLP

hidden dimension $d_M$. The parameterization transforms evolutionary operators into stochastic yet differentiable functions, where structured randomness maintains exploration without compromising gradient-based adaptation.

## 3.3 ADAPTIVE PARAMETER LEARNING

As shown in Fig. 2 (Top), ABOM's optimization loop comprises: 1) **Initialization**: The initial population $\mathbf{P}^{(0)}$ is randomly generated by Latin hypercube sampling; 2) **Reproduction**: Offspring $\hat{\mathbf{P}}^{(t)}$ are generated via $\hat{\mathbf{P}}^{(t)} = \pi_{\boldsymbol{\theta}}(\mathbf{P}^{(t)}, \mathbf{F}^{(t)})$; 3) **Evaluation**: Fitness values $\hat{\mathbf{F}}^{(t)}$ are computed for $\hat{\mathbf{P}}^{(t)}$; 4) **Elitism** Deb et al. (2002): The elite archive $\mathbf{E}^{(t)} \in \mathbb{R}^{N \times d}$, formed by the top $N$ individuals from $\mathbf{P}^{(t)} \cup \hat{\mathbf{P}}^{(t)}$, and their fitness values $\mathbf{F}_{\mathbf{E}}^{(t)} \in \mathbb{R}^N$, are carried over to the next generation; 5) **Parameter adaptation**: $\boldsymbol{\theta}$ is updated via adaptive parameter learning. The pseudocode of ABOM can be found in the Appendix C (Alg. 1). Crucially, ABOM performs adaptive parameter learning by minimizing the distance between offspring and the elite archive:

$$\min_{\boldsymbol{\theta}} \mathcal{L}^{(t)} = \|\hat{\mathbf{P}}^{(t)} - \mathbf{E}^{(t)}\|^2, \tag{10}$$

where, $\mathbf{E}^{(t)}$ denotes the elite archive. The objective refines evolutionary operators using task-specific knowledge from $\mathcal{M}^{(t)}$. From a learning perspective, adaptive parameter learning operates in a supervised paradigm. Gradients of $\mathcal{L}$ with respect to $\boldsymbol{\theta}$ are computed, and $\boldsymbol{\theta} \leftarrow \boldsymbol{\theta} - \eta \nabla_{\boldsymbol{\theta}} \mathcal{L}^{(t)}$ is updated via a gradient-based optimizer (e.g., AdamW Loshchilov & Hutter (2019)). The process ensures continuous adaptation to the target task without handcrafted training tasks.

**Discussion.** ABOM introduces three algorithmic properties that enhance its suitability for BBO: (1) **Learnable operators**: evolutionary mechanisms are parameterized and adapted online via gradient-based learning, reducing reliance on hand-designed heuristics; (2) **GPU-parallelizable design**: neural computation enables efficient batched execution on GPU, reducing wall-clock time per iteration; and (3) **Interpretable dynamics**: learned selection and mutation matrices reveal structured patterns in solution-fitness interactions and dimensional dependencies.

## 3.4 COMPUTATIONAL COMPLEXITY AND CONVERGENCE ANALYSIS

The computational cost of ABOM is primarily dominated by the selection, crossover, and mutation. The selection matrix (Eq. 5) incurs complexity $O(Ndd_A + N^2 d_A)$, where $N$ is the population size, $d$ the search space dimension, and $d_A$ the attention dimension. The MLP of the crossover (Eq. 6) contributes $O(Nd_A d_M + Nd_M d)$, with $d_M$ the hidden dimension of the MLP. The mutation (Eq. 7) contributes $O(d^2 d_A + dd_A d_M)$. Summing these, the total complexity is:

$$O(Ndd_A + N^2 d_A + Nd_A d_M + Nd_M d + d^2 d_A + dd_A d_M). \tag{11}$$

Assuming $d_A = d_M = d$ for simplicity, the formulation (11) reduces to $O(Nd^2 + N^2 d + d^3)$. In typical high-dimensional optimization ($N \ll d$), the leading term is $O(d^3)$, indicating that computational cost is primarily governed by the problem dimension. Note that $d_A$ and $d_M$ can be adjusted in practice to balance expressivity and efficiency. Next, we establish that ABOM achieves global convergence under the following assumption:

**Assumption 1** *The search space $\mathcal{X} \subseteq \mathbb{R}^d$ is compact, the objective $f_T$ is continuous with global minimizer $\mathbf{x}^*$ in the interior of $\mathcal{X}$, and ABOM uses $\tanh$-activated MLPs ($d_M \geq 1$) with dropout rates ($0 < p_C, p_M < 1$) during inference (operator execution).*

Let $f_t^* = \min_{\mathbf{x} \in \mathbf{E}^{(t)}} f_T(\mathbf{x})$ denote the best objective value in the elite archive. The filtration $\mathcal{F}_t = \sigma(\mathbf{P}^{(0)}, \ldots, \mathbf{P}^{(t)}, \theta^{(0)}, \ldots, \theta^{(t)})$ captures all algorithmic history up to generation $t$. ABOM preserves a non-vanishing probability of generating offspring $\hat{\mathbf{p}}_i^{(t)}$ near the global optimum:

**Corollary 1 (Exploration Guarantee)** *For any $\delta > 0$, $\exists \gamma > 0$ such that $\forall t \geq 0$,*

$$\mathbb{P}\big(\exists i : \|\hat{\mathbf{p}}_i^{(t)} - \mathbf{x}^*\| < \delta \mid \mathcal{F}_t\big) \geq 1 - (1 - \gamma)^N > 0. \tag{12}$$

Let $f^* = f_T(\mathbf{x}^*)$ be the global optimum value. Corollary 2 establishes a positive drift condition: when $f_t^* > f^* + \epsilon$, the expected improvement is strictly positive.

**Corollary 2 (Progress Guarantee)** *For any $\epsilon > 0$, $\exists \eta(\epsilon) > 0$ such that $\forall t \geq 0$,*

$$\mathbb{E}\big[f_t^* - f_{t+1}^* \mid \mathcal{F}_t,\, f_t^* > f^* + \epsilon\big] \geq \eta(\epsilon). \tag{13}$$

Combining these properties, we have:

**Theorem 3.1 (Global Convergence)** *Under Assumption 1, ABOM converges to the global optimum almost surely:*

$$f_t^* \xrightarrow{a.s.} f^* \quad as \quad t \to \infty. \tag{14}$$

*All proofs are provided in Appendix D.*

## 4 EXPERIMENTS

In this section, we address the following research questions: RQ1 (Performance Comparison): How does ABOM compare against classical and state-of-the-art BBO baselines on both synthetic and real-world benchmarks? RQ2 (Visualization Study): What statistical patterns emerge in ABOM's selection and mutation matrices? RQ3 (Ablation Study): Are all components of ABOM necessary for achieving competitive performance? RQ4 (Parameter Analysis): How sensitive is ABOM's performance to its key hyperparameters? We first describe the experimental setup and then systematically address RQ1–RQ4.

### 4.1 EXPERIMENTAL SETUP

**BBO Tasks**. We evaluate ABOM on the advanced MetaBox Benchmark Ma et al. (2023; 2025a), comprising both the synthetic black-box optimization benchmark (BBOB) Hansen et al. (2021) and the realistic unmanned aerial vehicle (UAV) path planning benchmark Shehadeh & Küdela (2025). The BBOB benchmark suite, widely adopted for evaluating black-box optimizers, comprises 24 continuous functions that exhibit diverse global optimization characteristics, including unimodal, multimodal, rotated, and shifted structures, with varying properties of Lipschitz continuity and second-order differentiability. We set the search space to $[-100, 100]^d$ with $d = 30/100/500$. The UAV benchmark provides 56 terrain-based problem instances for path planning in realistic landscapes with cylindrical threats. The objective is to select a specified number of path nodes in 3D space to minimize the total flight path length while ensuring collision-free navigation. The maximum function evaluations for BBOB and UAV are set to 20,000 and 2,500, respectively. All experiments are conducted on a Linux platform with an NVIDIA RTX 2080 Ti GPU (12 GB memory, CUDA 11.3). Detailed task configurations and other experimental results are provided in Appendices F, H, and J.

**Baselines**. We compare ABOM against three categories of baselines: (1) **Traditional BBO methods**: Random Search (RS) Bergstra & Bengio (2012), PSO Kennedy & Eberhart (1995), and DE Storn & Price (1997); (2) **Adaptive optimization variants**: SAHLPSO (advanced adaptive PSO variant) Tao et al. (2021), JDE21 (advanced adaptive DE variant) Brest et al. (2021), and CMAES (state-of-the-art adaptive ES variant) Hansen (2016); Ollivier et al. (2017); (3) **MetaBBO methods**: GLEET (advanced MetaBBO for PSO) Ma et al. (2024), RLDEAFL (advanced MetaBBO for DE) Guo et al. (2025), LES (advanced MetaBBO for ES) Lange et al. (2023b), and GLHF (advanced MetaBBO for solution manipulation) Li et al. (2024). All baselines follow the configurations outlined in the original papers. For all MetaBBO methods, we train them in the same problem distribution as RLDEAFL Guo et al. (2025) using the recommended settings. For BBOB, 8 out of the 24 problem instances are used as the training set, and the remaining 16 instances ($f_4, f_6 \sim f_{14}, f_{18} \sim f_{20}, f_{22} \sim f_{24}$) serve as the test set. For UAV, the 56 problem instances are evenly divided into training and test sets, with a partition of 50% / 50%. All parameter configurations are provided in the Appendix G.

### 4.2 PERFORMANCE COMPARISON (RQ1)

**Results on BBOB**. We evaluate ABOM against the baselines on the BBOB suite with $d = 30/100/500$. Tabless 1, 7, and 8 (See Appendix K) show the mean and standard deviation over

Table 1: The comparison results of the baselines on the BBOB suite with $d = 500$. All results are reported as the mean and standard deviation (mean $\pm$ std) over 30 independent runs. Symbols "$-$", "$\approx$", and "$+$" imply that the corresponding baseline is significantly worse, similar, and better than ABOM on the Wilcoxon rank-sum test with 95% confidence level, respectively. The best results are indicated in **bold**, and the suboptimal results are underlined.

| ID | Traditional BBO | | | Adaptive Variants | | | MetaBBO | | | | Ours |
|---|---|---|---|---|---|---|---|---|---|---|---|
| | RS | PSO | DE | SAHLPSO | JDE21 | CMAES | GLEET | RLDEAFL | LES | GLHF | ABOM |
| $f_4$ | 3.700e+5 ±1.192e+4 | 8.863e+4 ±1.525e+4 | 3.166e+5 ±3.506e+4 | 2.947e+5 ±2.418e+4 | 7.876e+4 ±2.086e+4 | 1.447e+4 ±7.83e+2 | 2.605e+5 ±2.173e+4 | 4.573e+4 ±1.079e+4 | 2.363e+5 ±4.075e+3 | 2.324e+5 ±7.512e+3 | **1.215e+4** ±**5.389e+2** |
| $f_6$ | 1.529e+7 ±6.327e+5 | 5.266e+6 ±5.262e+5 | 1.206e+7 ±1.390e+6 | 1.026e+7 ±3.e+5 | 4.399e+6 ±9.051e+5 | 2.164e+4 ±6.814e+3 | 9.870e+6 ±2.409e+5 | 1.875e+6 ±2.855e+5 | 9.253e+6 ±9.110e+4 | 9.194e+6 ±2.104e+5 | **6.201e+3** ±**6.326e+2** |
| $f_7$ | 3.92e+4 ±1.094e+3 | 2.811e+4 ±2.309e+3 | 3.366e+4 ±2.93e+3 | 2.774e+4 ±2.246e+3 | 1.856e+4 ±2.522e+3 | 1.289e+5 ±5.883e+2 | 2.525e+4 ±1.623e+3 | 1.481e+4 ±1.657e+3 | 2.285e+4 ±2.143e+2 | 2.073e+4 ±3.669e+2 | **2.432e+3** ±**2.250e+2** |
| $f_8$ | 6.052e+8 ±2.354e+7 | 4.083e+8 ±3.847e+7 | 4.153e+8 ±4.952e+7 | 2.194e+8 ±2.342e+7 | 1.332e+8 ±2.237e+7 | 2.827e+5 ±6.292e+4 | 1.183e+8 ±1.271e+7 | 5.807e+7 ±1.28e+7 | 5.068e+7 ±2.664e+5 | 5.055e+7 ±5.094e+5 | **8.886e+4** ±**1.267e+5** |
| $f_9$ | 4.159e+8 ±1.368e+7 | 1.719e+8 ±2.766e+7 | 2.002e+8 ±3.504e+7 | 5.151e+7 ±1.06e+7 | 3.326e+7 ±1.238e+7 | 2.533e+5 ±5.445e+4 | 1.473e+7 ±3.238e+6 | 1.145e+7 ±3.538e+6 | 4.548e+3 ±3.713e+0 | **3.243e+3** ±**5.560e-2** | 1.792e+5 ±5.876e+4 |
| $f_{10}$ | 2.832e+8 ±1.366e+7 | 8.026e+7 ±9.438e+6 | 2.440e+8 ±3.217e+7 | 2.229e+8 ±2.573e+7 | 5.459e+7 ±1.667e+7 | 1.580e+7 ±2.835e+6 | 2.097e+8 ±2.18e+7 | 2.232e+7 ±3.291e+6 | 2.085e+8 ±4.324e+6 | 2.002e+8 ±9.454e+6 | **5.958e+6** ±**5.916e+5** |
| $f_{11}$ | 5.881e+3 ±1.591e+2 | 6.187e+3 ±7.026e+2 | 4.999e+3 ±5.091e+2 | 4.835e+3 ±8.117e+2 | 4.371e+3 ±5.691e+2 | 1.248e+4 ±2.251e+2 | 3.722e+3 ±3.575e+2 | 3.259e+3 ±2.521e+2 | 5.107e+3 ±8.481e+1 | **2.529e+3** ±**3.248e+1** | 5.392e+3 ±3.159e+2 |
| $f_{12}$ | 3.015e+10 ±2.064e+9 | 1.414e+10 ±1.401e+9 | 2.055e+10 ±4.629e+9 | 1.675e+10 ±3.634e+9 | 4.800e+9 ±9.841e+8 | 1.32e+8 ±2.254e+7 | 1.235e+10 ±2.046e+9 | 2.819e+9 ±3.899e+8 | 1.084e+10 ±5.738e+8 | 9.757e+9 ±6.844e+8 | **2.733e+7** ±**4.903e+7** |
| $f_{13}$ | 1.444e+4 ±1.618e+2 | 1.319e+4 ±2.75e+2 | 1.324e+4 ±4.159e+2 | 1.197e+4 ±2.857e+2 | 8.994e+3 ±6.370e+2 | 2.363e+3 ±1.43e+2 | 1.116e+4 ±3.338e+2 | 7.073e+3 ±3.424e+2 | 1.255e+4 ±5.007e+1 | 1.024e+4 ±5.299e+1 | **1.221e+3** ±**3.010e+2** |
| $f_{14}$ | 6.634e+2 ±2.241e+1 | 4.824e+2 ±4.296e+1 | 5.081e+2 ±7.419e+1 | 4.208e+2 ±4.277e+1 | 1.842e+2 ±2.998e+1 | 2.494e+1 ±3.554e+0 | 3.231e+2 ±2.764e+1 | 1.157e+2 ±1.216e+1 | 1.458e+3 ±6.108e+0 | 2.542e+2 ±8.222e+0 | **1.487e+1** ±**2.291e+0** |
| $f_{18}$ | 1.428e+2 ±5.025e+0 | 1.017e+2 ±9.614e+0 | 1.093e+2 ±8.297e+0 | 9.814e+1 ±7.383e+0 | 7.074e+1 ±8.102e+0 | 1.100e+3 ±1.069e+1 | 8.539e+1 ±2.764e+0 | 5.495e+1 ±4.191e+0 | 3.704e+2 ±6.062e-1 | 6.875e+1 ±1.422e+0 | **3.792e+1** ±**4.075e+0** |
| $f_{19}$ | 2.09e+3 ±6.041e+1 | 9.012e+2 ±1.148e+2 | 9.606e+2 ±1.704e+2 | 2.764e+2 ±4.896e+1 | 1.772e+2 ±5.848e+1 | 1.374e+1 ±5.336e-1 | 8.479e+1 ±1.668e+1 | 8.767e+1 ±2.029e+1 | 2.502e+3 ±8.225e-1 | **2.504e-1** ±**3.113e-6** | 1.813e+1 ±1.603e+0 |
| $f_{20}$ | 3.233e+6 ±1.022e+5 | 2.374e+6 ±2.320e+5 | 2.432e+6 ±2.800e+5 | 1.471e+6 ±1.945e+5 | 5.884e+5 ±1.564e+5 | 3.802e+3 ±1.702e+3 | 9.466e+5 ±1.069e+5 | 2.136e+5 ±6.086e+4 | 3.772e+5 ±6.088e+3 | 3.753e+5 ±6.417e+3 | 2.565e+2 ±**1.351e+3** |
| $f_{22}$ | 8.636e+1 ±1.942e-2 | 8.609e+1 ±9.432e-2 | 8.61e+1 ±1.371e-1 | 8.542e+1 ±2.480e-1 | 8.003e+1 ±1.838e+0 | 2.851e+1 ±3.852e-1 | 8.478e+1 ±2.648e-1 | 7.159e+1 ±2.18e+0 | 1.184e+3 ±4.394e-2 | 8.356e+1 ±7.540e-2 | **4.971e+0** ±**6.520e+0** |
| $f_{23}$ | 1.652e+0 ±3.551e-2 | 1.641e+0 ±4.291e-2 | 1.659e+0 ±2.456e-2 | 1.658e+0 ±5.790e-2 | 1.586e+0 ±6.784e-2 | **3.959e-1** ±**3.54e-2** | 1.659e+0 ±4.429e-2 | 1.559e+0 ±1.525e-1 | 1.202e+3 ±3.532e-2 | 1.663e+0 ±3.539e-2 | 1.656e+0 ±3.434e-2 |
| $f_{24}$ | 2.089e+4 ±2.555e+2 | 1.604e+4 ±5.679e+2 | 1.610e+4 ±9.073e+2 | 1.222e+4 ±4.744e+2 | 1.198e+4 ±9.584e+2 | 4.986e+4 ±3.337e+1 | 1.010e+4 ±4.16e+2 | 9.9e+3 ±6.456e+2 | 8.822e+3 ±5.392e+1 | **7.437e+3** ±**7.572e+1** | 8.09e+3 ±3.268e+2 |
| $-/\approx/+$ | 15/1/0 | 15/1/0 | 14/1/1 | 14/1/1 | 14/1/1 | 15/0/1 | 14/1/1 | 14/1/1 | 14/0/2 | 11/1/4 | - |

30 runs for each baseline. Convergence curves of average normalized cost across all cases are provided in the Appendix K. ABOM matches or outperforms all baselines, achieving state-of-the-art performance, which validates the effectiveness of the proposed method. Both ABOM and adaptive optimization methods adjust the parameters online using optimization data. ABOM's parameterized operators offer greater flexibility than the fixed adaptation rules in the variant, leading to stronger performance. Compared to existing metaBBO algorithms, ABOM's improvements highlight the importance of parameter adaptation in enabling effective meta-optimization across diverse problem instances. The results suggest that adaptive mechanisms can support competitive performance without relying on handcrafted training tasks.

**Results on UAV**. We evaluate ABOM on 28 UAV benchmarks to validate its practical effectiveness. Fig. 3 shows the convergence of the normalized cost and runtime. ABOM converges fastest under limited evaluations and achieves the lowest normalized cost. Unlike metaBBO-based methods (e.g., GLHF, GLEET, RLDEAFL, LES), ABOM and adaptive optimization methods eliminate the need for training on hand-crafted tasks and associated overhead. Through GPU-accelerated evolution and adaptive parameter learning, ABOM achieves significantly faster runtime than most baselines.

## 4.3 Visualization Study (RQ2) and Ablation Study (RQ3)

We visualize the learned selection and mutation matrices of ABOM on three BBOB functions ($f_4$, $f_{11}$, $f_{24}$, $d = 30$) in Fig. 4. In all cases, the matrices develop structured statistical patterns as optimization proceeds. The selection matrix shows row similarity, indicating that ABOM learns to generate offspring from a small subset of individuals. This behavior resembles the difference vector mechanism in DE Storn & Price (1997), which reflects the strong expressive capacity of the learnable operator. Individuals with higher fitness are preferentially selected, in line with the

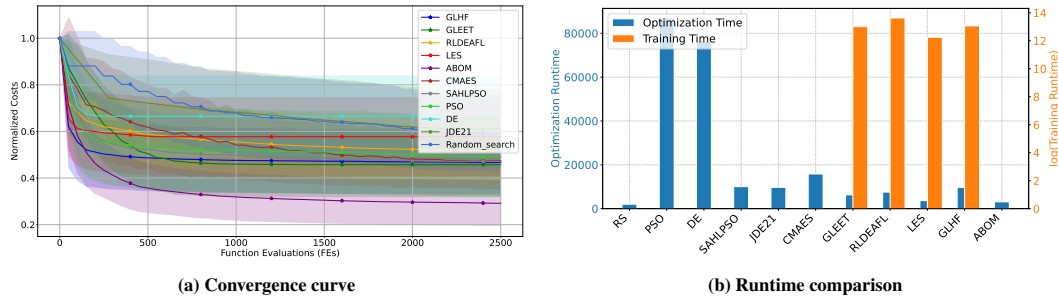

| (a) Convergence curve | (b) Runtime comparison |

Figure 3: Performance on 28 UAV problems: (Left) Convergence curve of average normalized cost across all problems. Costs (lower is better) are min-max normalized for each case. Detailed results are shown in the Appendix K; (Right) Average runtime (GPU seconds) over 30 independent runs.

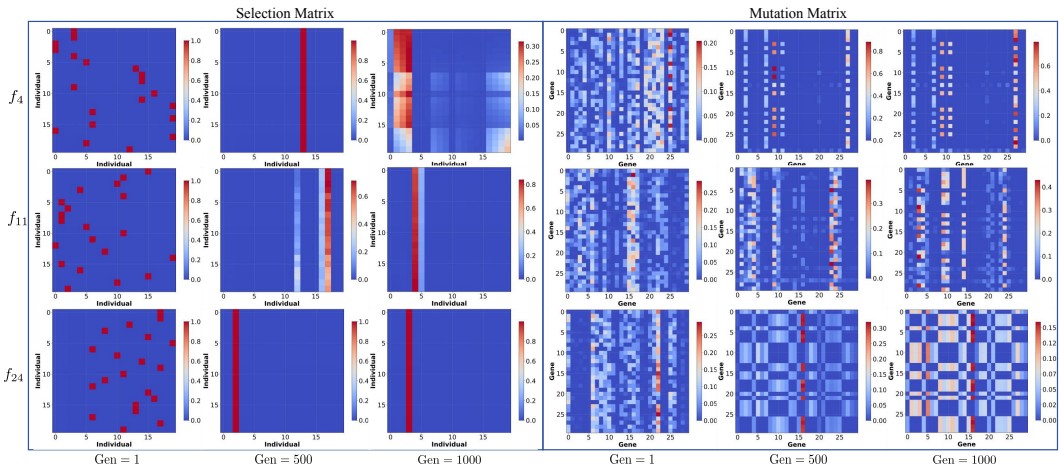

Figure 4: Learned selection and mutation matrices of ABOM on BBOB functions $f_4$, $f_{11}$, and $f_{24}$ ($d = 30$) at Generation 1, 500, and 1000. For the selection matrix, axes represent individuals ranked by their fitness values (0 is the best). For the mutation matrix, axes represent gene (variable) indices.

principle of survival of the fittest. The best individual is not always selected, which may help preserve population diversity. The mutation matrix evolves from random initialization to an ordered structure, suggesting that mutation follows consistent patterns adapted to the problem. These results demonstrate that ABOM provides greater interpretability than the metaBBO methods, which directly map neural networks to solution manipulation Li et al. (2024; 2025).

Table 2: Ablation study of ABOM's key components on the BBOB suite with $d = 30$.

| ID | No Crossover (mean ± std) | No Mutation (mean ± std) | No Parameter Adaptation (mean ± std) | **ABOM** (mean ± std) |
|---|---|---|---|---|
| $f_4$ | 4.23e+03 ± 3.02e+03 | 1.01e+03 ± 5.44e+02 | 2.58e+04 ± 1.67e+04 | **5.45e+02 ± 2.95e+02** |
| $f_6$ | 1.62e+04 ± 1.55e+04 | 1.10e+04 ± 1.95e+04 | 4.54e+04 ± 3.61e+04 | **2.60e+02 ± 2.64e+02** |
| $f_7$ | 6.39e+03 ± 6.91e+03 | 1.18e+04 ± 6.86e+02 | 1.67e+04 ± 1.11e+04 | **5.58e+02 ± 2.77e+02** |
| $f_8$ | 1.52e+03 ± 2.91e+03 | 1.94e+03 ± 3.62e+03 | 1.03e+08 ± 2.68e+08 | **1.15e+02 ± 1.56e+02** |
| $f_9$ | 1.96e+04 ± 7.02e+04 | 1.13e+03 ± 3.15e+03 | 2.49e+06 ± 5.94e+06 | **2.35e+03 ± 5.30e+03** |
| $f_{10}$ | 1.16e+07 ± 7.11e+06 | 1.07e+07 ± 3.48e+06 | 3.65e+07 ± 1.64e+07 | **9.72e+05 ± 7.38e+05** |
| $f_{11}$ | 1.05e+05 ± 3.19e+04 | 1.00e+05 ± 2.68e+04 | 8.00e+04 ± 2.20e+04 | **2.61e+04 ± 1.01e+04** |
| $f_{12}$ | 1.12e+09 ± 3.12e+09 | 2.30e+07 ± 5.74e+07 | 1.63e+10 ± 1.61e+10 | **5.28e+07 ± 1.45e+08** |
| $f_{13}$ | 7.54e+01 ± 4.06e+01 | 7.71e+01 ± 3.93e+01 | 8.71e+03 ± 3.16e+03 | **7.28e+01 ± 3.07e+01** |
| $f_{14}$ | 8.43e+01 ± 7.23e+01 | 9.29e+00 ± 2.49e+01 | 9.28e+02 ± 5.96e+02 | **3.46e-02 ± 3.39e-02** |
| $f_{18}$ | 1.18e+03 ± 2.21e+03 | 7.02e+02 ± 2.46e+02 | 4.94e+02 ± 1.66e+02 | **5.12e+02 ± 1.37e+02** |
| $f_{19}$ | 2.35e+01 ± 3.45e+01 | 1.23e+01 ± 1.23e+01 | 1.64e+02 ± 3.27e+02 | **2.48e-01 ± 1.11e-03** |
| $f_{20}$ | -6.54e+01 ± 4.95e+00 | -6.58e+01 ± 3.53e+00 | -5.82e+01 ± 4.19e+00 | **-6.57e+01 ± 3.80e+00** |
| $f_{22}$ | 8.66e+01 ± 0.00e+00 | 8.66e+01 ± 0.00e+00 | 8.66e+01 ± 0.00e+00 | **8.66e+01 ± 0.00e+00** |
| $f_{23}$ | 3.03e+00 ± 5.33e-01 | 3.12e+00 ± 4.54e-01 | 3.20e+00 ± 4.25e-01 | **3.01e-01 ± 2.19e-01** |
| $f_{24}$ | 2.92e+02 ± 2.46e+02 | 2.30e+02 ± 5.31e+01 | 4.94e+03 ± 3.41e+03 | **2.44e+02 ± 2.37e+01** |
| $-/ ≈/+$ | 13/3/0 | 10/3/3 | 14/1/1 | – |

We conduct an ablation study comparing the proposed ABOM with variants that disable specific mechanisms, including no crossover, no mutation, and no parameter adaptation. Table 2 presents the mean and standard deviation over 30 runs on the BBOB suite with $d = 30$. Table 2 illustrates that both crossover and mutation are crucial components, as their removal individually causes significant performance deterioration. Furthermore, the variant without parameter adaptation performs significantly worse than ABOM, underscoring the critical importance of the adaptive mechanism for achieving robust and high-quality optimization.

## 4.4 PARAMETER ANALYSIS (RQ4)

Fig. 5 illustrates ABOM's hyperparameter sensitivity on the BBOB suite ($d = 30$). A population size of 20 proves sufficient for robust performance across most functions within 20,000 evaluations. Similarly, a hidden dimension $d_M$ smaller than $d$ (e.g., $d_M = 16$) often achieves competitive results. In practice, $d_M$ should be carefully configured to balance computational efficiency and optimization quality effectively. The parameters $p_C$ and $p_M$ exhibit optimal performance at 0.95, indicating that higher values increase stochasticity and exploration. Nevertheless, setting either parameter to 1 eliminates beneficial randomness, degrading performance. Thus, controlled stochasticity is crucial for maintaining the balance between exploitation and diversity.

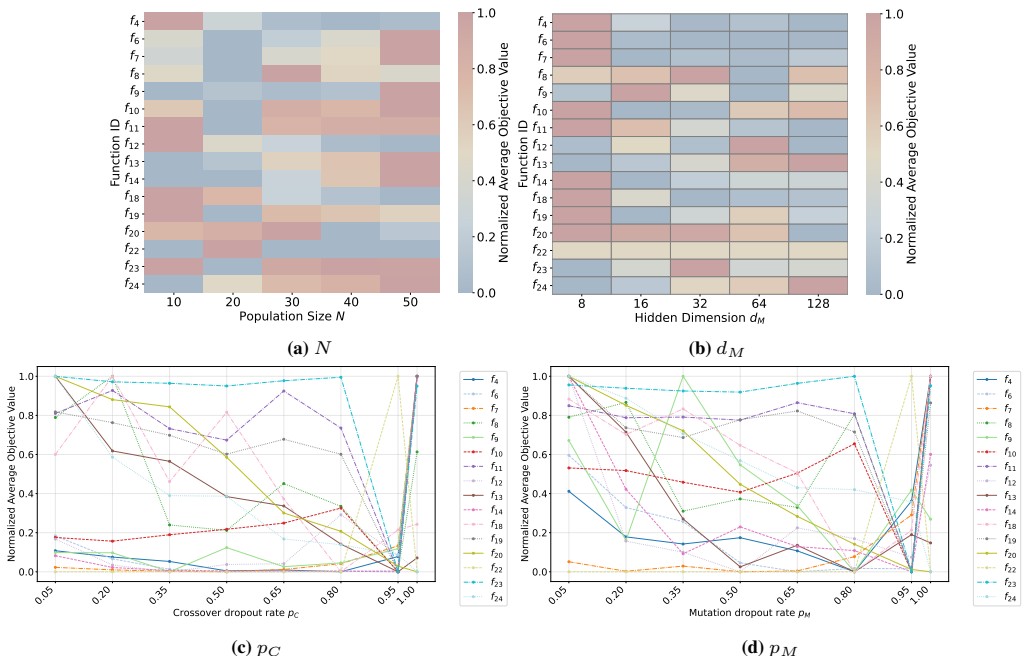

Figure 5: Sensitivity analysis of key hyperparameters on the BBOB suite with $d = 30$: Algorithm performance across different settings for population size ($N$), hidden dimension ($d_M$), crossover dropout rate ($p_C$), and mutation dropout rate ($p_M$). The learning rate analysis is in Appendix I.

## 5 CONCLUSION AND DISCUSSION

**Summary**. We present a task-free adaptive metaBBO method, ABOM, which eliminates dependency on handcrafted training tasks by performing online parameter adaptation using only optimization data from the target task. Unlike conventional MetaBBO methods that require offline meta-training, ABOM integrates parameter learning directly into the evolutionary loop, enabling zero-shot generalization. Our framework parameterizes evolutionary operators as differentiable modules, updated via gradient descent to align offspring with elites, thereby eliminating the need for pretraining or heuristic design. Empirical results in synthetic and realistic benchmarks demonstrate that ABOM matches or outperforms advanced baselines without prior task knowledge. Attention visualization reveals interpretable search behaviors with consistent structural patterns. Thus, ABOM establishes a task-free paradigm for metaBBO, where learning and search co-evolve in real time.

**Limitations and Future Work**. Current limitations motivate several promising directions: (1) Addressing the cubic computational bottleneck ($O(d^3)$) through sparse or low-rank attention mechanisms to reduce ABOM's complexity; (2) Dynamically adapting population size and model capacity during optimization; (3) Conducting a convergence rate analysis grounded in the theoretical examination of adaptive parameter learning in ABOM; and (4) Exploring hybrid training paradigms that integrate pretraining on prior knowledge with online adaptation, thereby enhancing optimization efficiency and bridging the gap between task-agnostic adaptation and cross-task generalization.

ACKNOWLEDGMENTS

This work was supported in part by the National Natural Science Foundation of China(No.62576264), Project supported by the National Science and Technology Major Project of the Ministry of Science and Technology of China (No.2025ZD0551500, No.2025ZD0551502) , the Key Project of National Natural Science Foundation of China (62431020,62231027), the Joint Fund Project of National Natural Science Foundation of China (No.U22B2054), the Fund for Foreign Scholars in University Research and Teaching Programs (the 111 Project) (No.B07048), the Postdoctoral Fellowship Program of China Postdoctoral Science Foundation (CPSF) (No.GZC20232033), the Natural Science Basic Research Program of Shaanxi (Program No. 2025JC-YBQN-795), the China Postdoctoral Science Foundation (Certificate Number: 2025T180431 and 2025M771550), the Program for Cheung Kong Scholars and Innovative Research Team in University (No.IRT 15R53), the Key Scientific Technological Innovation Research Project by Ministry of Education and the National Key Laboratory of Human-Machine Hybrid Augmented Intelligence, Xi'an Jiaotong University (No.HMHAI-202404, No. HMHAI-202405).

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

## A    REPRODUCIBILITY STATEMENT

To ensure the reproducibility, we have taken the following measures:

- **Source Code**: We provide a complete implementation of the Adaptive meta Black-box Optimization Model (ABOM) at the following repository: `https://github.com/xiaofangxd/ABOM`.

- **Algorithm Specification**: ABOM is fully detailed in Section 3, including the mathematical formulations for the selection, crossover, and mutation operators. The complete pseudocode for the optimization loop is provided in Algorithm 1 (Appendix C).

- **Experimental Setup**: All baselines are implemented and evaluated using the state-of-the-art MetaBox benchmark platform Ma et al. (2023; 2025a). This ensures a standardized, fair, and reproducible comparison. Appendix F describes the BBOB and UAV benchmarks, including their characteristics, search space dimensions, and evaluation budgets. Appendix G details the hyperparameter settings for all baselines as specified in the original papers and implemented in MetaBox.

By providing the code, algorithmic descriptions, and experimental configurations on the unified MetaBox platform, we aim to enable other researchers to fully reproduce and build upon our results.

## B    USE OF LARGE LANGUAGE MODELS

Large Language Models (LLMs) were used solely as a general-purpose writing assistance tool. Their role was limited to language polishing, grammatical refinement, and improving the clarity and fluency of the paper. LLMs did not contribute to the conception of research ideas, experimental design, data analysis, or interpretation of results. All intellectual contributions, including the formulation of the problem, methodology, and conclusions, were made entirely by the human authors.

## C    PSEUDOCODE OF ABOM

---
**Algorithm 1** Adaptive meta Black-box Optimization Model (ABOM)

---
**Input:** Target black-box optimization task $f_T$, population size $N$, max generations $T$, crossover dropout rate $p_C$, mutation dropout rate $p_M$, learning rate $\eta$, attention dimension $d_A$, and MLP hidden dimension $d_M$.

1: Initialize $\mathbf{P}^{(0)}$ via Latin hypercube sampling;
2: Evaluate: $\mathbf{F}^{(0)} \leftarrow f_T(\mathbf{P}^{(0)})$;
3: **for** $t = 0$ **to** $T - 1$ **do**
4:     Generate offspring: $\hat{\mathbf{P}}^{(t)} \leftarrow \pi_{\boldsymbol{\theta}}(\mathbf{P}^{(t)}, \mathbf{F}^{(t)}; d_A, d_M, p_C, p_M)$ ;
5:     Evaluate: $\hat{\mathbf{F}}^{(t)} \leftarrow f_T(\hat{\mathbf{P}}^{(t)})$;
6:     Form elite archive: $\mathbf{E}^{(t)}, \mathbf{F}_{\mathbf{E}}^{(t)} \leftarrow \text{top}_N\left(\mathbf{P}^{(t)} \cup \hat{\mathbf{P}}^{(t)}\right), \text{top}_N\left(\mathbf{F}^{(t)} \cup \hat{\mathbf{F}}^{(t)}\right)$;
7:     Update parameters by AdamW: $\boldsymbol{\theta} \leftarrow \boldsymbol{\theta} - \eta \nabla_{\boldsymbol{\theta}} \left\|\hat{\mathbf{P}}^{(t)} - \mathbf{E}^{(t)}\right\|^2$;
8:     Elitism: $\mathbf{P}^{(t+1)} \leftarrow \mathbf{E}^{(t)}, \mathbf{F}^{(t+1)} \leftarrow \mathbf{F}_{\mathbf{E}}^{(t)}$;
9: **end for**
**Output:** Optimal individual (solution) $\mathbf{p}^* = \arg\min_{\mathbf{p} \in \mathbf{P}^{(t)}} f_T(\mathbf{p})$.

---

## D    CONVERGENCE ANALYSIS OF ABOM

This section presents a convergence analysis of ABOM under some assumptions. We rigorously prove that ABOM converges with probability 1 to the global optimum of the objective function. Let $\mathcal{X} \subseteq \mathbb{R}^d$ be a compact search space and $f_T : \mathcal{X} \to \mathbb{R}$ be a continuous objective function with global minimum $f^* = f_T(\mathbf{x}^*)$. ABOM maintains a population $\mathbf{P}^{(t)} \in \mathbb{R}^{N \times d}$ at generation

$t$, with corresponding fitness values $\mathbf{F}^{(t)}$. For the convergence analysis, we make the following assumptions:

**Assumption 2** *The following conditions hold:*

  *(i) The global optimum $\mathbf{x}^*$ lies in the interior of $\mathcal{X}$.*

  *(ii) Dropout rates satisfy $0 < p_C, p_M < 1$.*

  *(iii) MLP hidden dimension $d_M \geq 1$ with tanh activation.*

Define $f_t^* = \min_{\mathbf{x} \in \mathbf{E}^{(t)}} f_T(\mathbf{x})$, where $\mathbf{E}^{(t)}$ is the elite archive containing the top $N$ individuals from $\mathbf{P}^{(t)} \cup \hat{\mathbf{P}}^{(t)}$. Let $\mathcal{F}_t = \sigma(\mathbf{P}^{(0)}, \ldots, \mathbf{P}^{(t)}, \theta^{(0)}, \ldots, \theta^{(t)})$ be the filtration representing all information up to generation $t$. The elitism mechanism ensures $f_{t+1}^* \leq f_t^*$ almost surely, implying $\mathbb{E}[f_{t+1}^* \mid \mathcal{F}_t] \leq f_t^*$. Since $f_T$ is bounded on the compact set $\mathcal{X}$, the sequence $\{f_t^*, \mathcal{F}_t\}$ forms a lower-bounded supermartingale. By the martingale convergence theorem Hall & Heyde (2014), $f_t^*$ converges with probability 1 to the random variable $f_\infty^* \geq f^*$. To establish global convergence, we need to prove $f_\infty^* = f^*$ with probability 1.

**Lemma 1** *Under Assumption 2, for any $\delta > 0$, there exists $\gamma > 0$ such that:*

$$\mathbb{P}(\exists i : \|\hat{\mathbf{p}}_i^{(t)} - \mathbf{x}^*\| < \delta \mid \mathcal{F}_t) \geq 1 - (1 - \gamma)^N > 0. \tag{15}$$

**Proof 1** *For the crossover operation, consider any parent $\mathbf{p}_i^{(t)} \in \mathcal{X}$ and let $\mathbf{v} = \mathbf{x}^* - \mathbf{p}_i^{(t)}$. Define the MLP configuration with $\mathbf{W}_1 = \mathbf{0}$, $\mathbf{b}_1 = \mathbf{0}$, $\mathbf{W}_2 = \mathbf{0}$, and $\mathbf{b}_2 = \mathbf{v}$. Then:*

$$\text{MLP}_{\theta_c^*}(\mathbf{A}^{(t)}\mathbf{p}_i^{(t)}) = \tanh(\mathbf{A}^{(t)}\mathbf{p}_i^{(t)}\mathbf{W}_1 + \mathbf{b}_1)\mathbf{W}_2 + \mathbf{b}_2 = \mathbf{v}. \tag{16}$$

*Consequently:*

$$\mathbf{p}_i^{(t)} + \text{MLP}_{\theta_c^*}(\mathbf{A}^{(t)}\mathbf{p}_i^{(t)}) = \mathbf{x}^*. \tag{17}$$

*By continuity of the MLP (as a composition of continuous functions), there exists $\epsilon > 0$ such that for all $\theta_c \in \mathcal{N}_\epsilon(\theta_c^*)$:*

$$\|\mathbf{p}_i^{(t)} + \text{MLP}_{\theta_c}(\mathbf{A}^{(t)}\mathbf{p}_i^{(t)}) - \mathbf{x}^*\| < \delta/2. \tag{18}$$

*Define $\mu_c = \mathbb{P}(\theta_c^{(t)} \in \mathcal{N}_\epsilon(\theta_c^*) \mid \mathcal{F}_t)$. Given the parameter update $\theta_c^{(t+1)} = \theta_c^{(t)} - \eta \nabla \mathcal{L}(\theta_c^{(t)}) + \xi^{(t)}$ with stochastic perturbations $\xi^{(t)}$ from dropout patterns $\mathbf{D}^{(t)} \sim \text{Bernoulli}(1 - p_C)^{d_M}$, which have minimum probability mass:*

$$\min_{\mathbf{D}} \mathbb{P}(\mathbf{D}^{(t)} = \mathbf{D}) = (\min\{p_C, 1 - p_C\})^{d_M} > 0, \tag{19}$$

*and since the conditional distribution of $\theta_c^{(t)}$ has positive density, there exists $c_t > 0$ such that:*

$$\mu_c \geq (\min\{p_C, 1 - p_C\})^{d_M} \cdot c_t > 0. \tag{20}$$

*For the mutation operation, consider any intermediate solution $\mathbf{p}_i^{'(t)}$ and let $\mathbf{w} = \mathbf{x}^* - \mathbf{p}_i^{'(t)}$. Define the MLP configuration with $\mathbf{W}_3 = \mathbf{0}$, $\mathbf{b}_3 = \mathbf{0}$, $\mathbf{W}_4 = \mathbf{0}$, and $\mathbf{b}_4 = \mathbf{w}$. Then:*

$$\text{MLP}_{\theta_m^*}(\mathbf{M}_i^{(t)}\mathbf{p}_i^{'(t)}) = \tanh(\mathbf{M}_i^{(t)}\mathbf{p}_i^{'(t)}\mathbf{W}_3 + \mathbf{b}_3)\mathbf{W}_4 + \mathbf{b}_4 = \mathbf{w}. \tag{21}$$

*Consequently:*

$$\mathbf{p}_i^{'(t)} + \text{MLP}_{\theta_m^*}(\mathbf{M}_i^{(t)}\mathbf{p}_i^{'(t)}) = \mathbf{x}^*. \tag{22}$$

*By identical continuity properties, there exists $\epsilon > 0$ such that for all $\theta_m \in \mathcal{N}_\epsilon(\theta_m^*)$:*

$$\|\mathbf{p}_i^{'(t)} + \text{MLP}_{\theta_m}(\mathbf{M}_i^{(t)}\mathbf{p}_i^{'(t)}) - \mathbf{x}^*\| < \delta/2. \tag{23}$$

*Define $\mu_m = \mathbb{P}(\theta_m^{(t)} \in \mathcal{N}_\epsilon(\theta_m^*) \mid \mathcal{F}_t)$. By analogous reasoning to the crossover operation:*

$$\mu_m \geq (\min\{p_M, 1 - p_M\})^{d_M} \cdot c_t' > 0, \tag{24}$$

where $c_t' > 0$ is determined by mutation parameters.

The probability that a single offspring $\hat{\mathbf{p}}_i^{(t)}$ falls within $\delta$ of $\mathbf{x}^*$ satisfies:

$$\mathbb{P}(\|\hat{\mathbf{p}}_i^{(t)} - \mathbf{x}^*\| < \delta \mid \mathcal{F}_t) \geq \mu_c \mu_m. \tag{25}$$

Let $\gamma = \mu_c \mu_m > 0$, which is a positive constant independent of $t$ and depends only on hyperparameters $p_C$, $p_M$, $d_M$, and the adaptive parameter learning. Finally, for $N$ independent offspring:

$$\mathbb{P}(\exists i : \|\hat{\mathbf{p}}_i^{(t)} - \mathbf{x}^*\| < \delta \mid \mathcal{F}_t) = 1 - \prod_{i=1}^{N}(1 - \mathbb{P}(\|\hat{\mathbf{p}}_i^{(t)} - \mathbf{x}^*\| < \delta \mid \mathcal{F}_t)) \geq 1 - (1 - \gamma)^N > 0. \tag{26}$$

Define the distance function $V_t = f_t^* - f^* \geq 0$. Using Lemma 1, we establish the following drift condition He & Yao (2001); Zhou et al. (2019):

**Lemma 2** *Under Assumption 2, for any $\epsilon > 0$, there exists $\eta(\epsilon) > 0$ such that:*

$$\mathbb{E}[V_t - V_{t+1} \mid \mathcal{F}_t, V_t > \epsilon] \geq \eta(\epsilon) > 0. \tag{27}$$

**Proof 2** *Define the event $A_t = \{\exists i : \|\hat{\mathbf{p}}_i^{(t)} - \mathbf{x}^*\| < \delta\}$, where $\delta > 0$ is chosen such that for all $\mathbf{x} \in \mathcal{X}$ with $\|\mathbf{x} - \mathbf{x}^*\| < \delta$, we have $f_T(\mathbf{x}) < f^* + \epsilon/2$ (which exists by the continuity of $f_T$ and Assumption 2(i)). By Lemma 1, there exists $\gamma > 0$ such that:*

$$\mathbb{P}(A_t \mid \mathcal{F}_t, V_t > \epsilon) \geq 1 - (1 - \gamma)^N > 0. \tag{28}$$

*When $A_t$ occurs, the elite archive at generation $t+1$ contains at least one solution with fitness value less than $f^* + \epsilon/2$, so:*

$$V_{t+1} = f_{t+1}^* - f^* < \epsilon/2. \tag{29}$$

*When $A_t$ does not occur, the elitism mechanism ensures $V_{t+1} \leq V_t$ (since the elite archive preserves the best solutions). Therefore, the expected drift can be decomposed as:*

$$\mathbb{E}[V_t - V_{t+1} \mid \mathcal{F}_t, V_t > \epsilon] = \mathbb{E}[V_t - V_{t+1} \mid \mathcal{F}_t, V_t > \epsilon, A_t] \cdot \mathbb{P}(A_t \mid \mathcal{F}_t, V_t > \epsilon) \tag{30}$$
$$+ \mathbb{E}[V_t - V_{t+1} \mid \mathcal{F}_t, V_t > \epsilon, A_t^c] \cdot \mathbb{P}(A_t^c \mid \mathcal{F}_t, V_t > \epsilon). \tag{31}$$

*For the first term, using Eq. (29) and the condition $V_t > \epsilon$:*

$$\mathbb{E}[V_t - V_{t+1} \mid \mathcal{F}_t, V_t > \epsilon, A_t] \geq V_t - \epsilon/2 \tag{32}$$
$$> \epsilon - \epsilon/2 = \epsilon/2. \tag{33}$$

*For the second term, since $V_{t+1} \leq V_t$ by the elitism mechanism:*

$$\mathbb{E}[V_t - V_{t+1} \mid \mathcal{F}_t, V_t > \epsilon, A_t^c] \geq 0. \tag{34}$$

*Combining these results with Eq. (28):*

$$\mathbb{E}[V_t - V_{t+1} \mid \mathcal{F}_t, V_t > \epsilon] \geq (\epsilon/2) \cdot (1 - (1 - \gamma)^N) + 0 \cdot \mathbb{P}(A_t^c \mid \mathcal{F}_t, V_t > \epsilon) \tag{35}$$
$$\geq (\epsilon/2) \cdot (1 - (1 - \gamma)^N). \tag{36}$$

*Setting $\eta(\epsilon) = (\epsilon/2) \cdot (1 - (1 - \gamma)^N) > 0$ completes the proof.*

With the positive drift condition established, we can prove global convergence.

**Theorem D.1** *Under Assumption 2, ABOM converges with probability 1 to the global optimum:*

$$f_t^* \xrightarrow{a.s.} f^* \quad as \quad t \to \infty. \tag{37}$$

**Proof 3** *By the martingale convergence theorem Hall & Heyde (2014), $f_t^*$ converges with probability 1 to some random variable $f_\infty^* \geq f^*$. Assume for contradiction that $f_\infty^* > f^*$ with positive*

*probability. Then there exists $\epsilon > 0$ such that $V_t > \epsilon$ for all sufficiently large $t$. Define the stopping time $\tau_k = \inf\{t \geq k : V_t \leq \epsilon\}$.*

*Consider the value function change from time $k$ to $\tau_k$:*

$$V_k - V_{\tau_k} = \sum_{t=k}^{\tau_k - 1} (V_t - V_{t+1}). \tag{38}$$

*Taking expectations and applying the law of iterated expectations:*

$$\mathbb{E}[V_k - V_{\tau_k}] = \mathbb{E}\left[ \sum_{t=k}^{\tau_k - 1} \mathbb{E}[V_t - V_{t+1} \mid \mathcal{F}_t] \right]. \tag{39}$$

*For $t < \tau_k$, we have $V_t > \epsilon$, so by Lemma 2:*

$$\mathbb{E}\left[ \sum_{t=k}^{\tau_k - 1} \mathbb{E}[V_t - V_{t+1} \mid \mathcal{F}_t] \right] \geq \eta(\epsilon) \cdot \mathbb{E}[\tau_k - k]. \tag{40}$$

*Thus:*

$$\mathbb{E}[V_k] - \mathbb{E}[V_{\tau_k}] \geq \eta(\epsilon) \cdot \mathbb{E}[\tau_k - k]. \tag{41}$$

*Since $V_{\tau_k} \leq \epsilon$, we have:*

$$\mathbb{E}[\tau_k - k] \leq \frac{\mathbb{E}[V_k]}{\eta(\epsilon)} < \infty. \tag{42}$$

*This implies $\mathbb{P}(\tau_k < \infty) = 1$, contradicting the assumption that $V_t > \epsilon$ for all sufficiently large $t$. Therefore, $f_\infty^* = f^*$ with probability 1.*

Theorem D.1 establishes that ABOM converges with probability 1 to the global optimum under Assumption 2. The persistent application of dropout during inference, coupled with the adaptive parameter learning mechanism, ensures that there exists a positive probability of generating offspring within an arbitrarily small neighborhood of the optimum at each iteration.

**Theoretical Limitation.** Theorem D.1 establishes asymptotic convergence but not polynomial-time convergence. Convergence rate analysis (expected hitting time) for specific problems is one of the important research directions for the future. It is worth noting that our convergence analysis does not directly apply in cases where the global optimum lies on the boundary or where constraint handling results in a discontinuous feasible region.

**Theoretical Contributions.** Our work establishes two theoretical contributions for meta black-box optimization: 1) We prove a novel exploration guarantee (Lemma 1) showing that attention-based MLP parameterization with dropout maintains persistent exploration capability. 2) We prove global convergence of ABOM with adaptive parameter learning (Theorem D.1). Crucially, this demonstrates that self-supervised parameter adaptation does not compromise convergence guarantees. This stands in contrast to existing neural network-parameterized methods such as GLHF Li et al. (2024) and B2Opt Li et al. (2025), which lack rigorous convergence analysis despite empirical success. These theoretical foundations provide ABOM's reliability while preserving the flexibility of adaptive optimization.

## E  CONVERGENCE ANALYSIS OF PARAMETER ADAPTATION

Fig. 6 shows the loss curves of parameter adaptation on the BBOB suite with $[-100, 100]^{500}$. Despite the large search space and high dimensionality, the loss of parameter adaptation consistently decreases and converges across all test functions, with minimal variance across 30 independent runs. This empirical evidence validates our theoretical assumption of local convergence for parameter adaptation (Eq. 18), demonstrating that the self-supervised learning paradigm with AdamW optimizer remains stable even in challenging high-dimensional optimization scenarios. The consistent convergence behavior aligns with standard machine learning practices for training attention-based MLP architectures using gradient-based optimization, confirming the practical viability of our adaptive parameter learning framework.

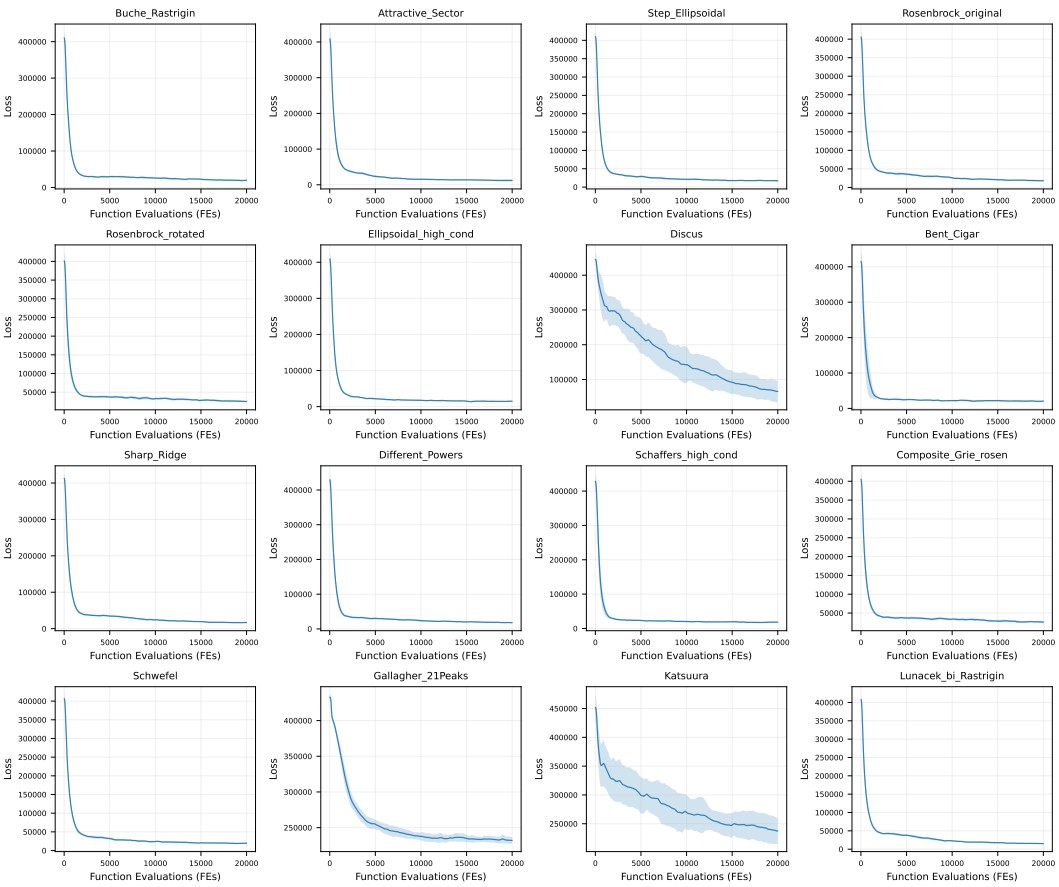

Figure 6: Loss curves of parameter adaptation on the BBOB suite with $d = 500$. Each subplot depicts the mean loss across 30 independent runs, with shaded regions representing standard deviation.

Fig. 7 presents the convergence behavior of ABOM and ABOM-NPA (no parameter adaptation). Both methods demonstrate convergence in practice, which empirically confirms that the convergence guarantee stems from the core mechanisms of elite preservation and dropout-enabled exploration rather than solely depending on parameter adaptation. While our theoretical analysis (Theorem D.1) formally establishes global convergence for ABOM with parameter adaptation, the experimental comparison reveals that the fundamental convergence properties are maintained even without this component, suggesting that the theoretical framework could be extended to cover variants without parameter adaptation. This empirical observation aligns with the martingale convergence argument in Theorem D.1, where the supermartingale property $\mathbb{E}[f_{t+1}^* | \mathcal{F}_t] \leq f_t^*$ is primarily ensured by the elitism mechanism rather than parameter adaptation.

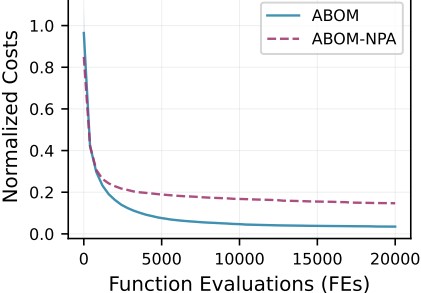

Figure 7: Convergence comparison between ABOM and ABOM-NPA on BBOB suite with $d = 500$, which shows normalized costs against function evaluations over 30 independent runs.

## F TASK CONFIGURATION

Table 3 presents 24 instances of synthetic black-box optimization benchmarks (BBOB) with diverse characteristics and landscapes. Following the standard protocol of the benchmark platform Ma et al. (2023; 2025a), functions $f_1$, $f_2$, $f_3$, $f_5$, $f_{15}$, $f_{16}$, $f_{17}$, and $f_{21}$ are designated as training

Table 3: Overview of the BBOB suites.

| ID | Function | Characteristic | Usage |
|----|----------|----------------|-------|
| $f_1$ | Sphere | | **Train** |
| $f_2$ | Ellipsoidal | Separable | **Train** |
| $f_3$ | Rastrigin | | **Train** |
| $f_5$ | Linear Slope | | **Train** |
| $f_{15}$ | Rastrigin (non-separable) | | **Train** |
| $f_{16}$ | Weierstrass | Multi-modal with adequate global structure | **Train** |
| $f_{17}$ | Schaffers F7 | | **Train** |
| $f_{21}$ | Gallagher's Gaussian 101-me Peaks | Multi-modal with weak global structure | **Train** |
| $f_4$ | Buche-Rastrigin | Separable | Test |
| $f_6$ | Attractive Sector | | Test |
| $f_7$ | Step Ellipsoidal | Low/moderate conditioning | Test |
| $f_8$ | Rosenbrock, original | | Test |
| $f_9$ | Rosenbrock, rotated | | Test |
| $f_{10}$ | Ellipsoidal | | Test |
| $f_{11}$ | Discus | | Test |
| $f_{12}$ | Bent Cigar | High conditioning, unimodal | Test |
| $f_{13}$ | Sharp Ridge | | Test |
| $f_{14}$ | Different Powers | | Test |
| $f_{18}$ | Schaffers F7, ill-conditioned | Multi-modal with adequate global structure | Test |
| $f_{19}$ | Composite Griewank-Rosenbrock F8F2 | | Test |
| $f_{20}$ | Schwefel | | Test |
| $f_{22}$ | Gallagher's Gaussian 21-hi Peaks | Multi-modal with weak global structure | Test |
| $f_{23}$ | Katsuura | | Test |
| $f_{24}$ | Lunacek bi-Rastrigin | | Test |

functions, while the remaining functions serve as test instances, ensuring a balanced distribution of optimization difficulty between the training and test sets. The maximum number of function evaluations is set to 20,000. All functions are defined over $[-100, 100]^d, d = 30/100/500$.

The UAV benchmarks comprise 56 terrain-based scenarios that represent realistic unmanned aerial vehicle path planning problems, each with 30 dimensions. The scenarios are divided into training and test sets of equal size (28 instances each), with test instances corresponding to even-numbered indices $(0, 2, 4, \ldots, 54)$. Following the standard protocol of the benchmark platform Ma et al. (2023; 2025a), the maximum number of function evaluations is set to 2,500.

## G  BASELINES

Since our ABOM is an evolution-based meta-black-box optimization (metaBBO) algorithm, we restrict comparisons exclusively to evolution-based methods, excluding non-evolution-based methods such as Bayesian optimization. Furthermore, we omit LLM-based metaBBO methods Liu et al. (2025); Romera-Paredes et al. (2024); Yang et al. (2024) from our baselines, as they are tailored for specific task types and are not directly comparable to evolution-based general-purpose frameworks.

To ensure a fair and reproducible comparison, all baselines are implemented using the source code provided by the official MetaBox platform Ma et al. (2023; 2025a). Detailed hyperparameter configurations for baselines are provided in Table 4, while the configuration of our proposed ABOM is summarized in Table 5. All results are reported as the mean and standard deviation over 30 independent runs, with a fixed population size of 20 across all trials.

For traditional BBO methods, we adopt the hyperparameter settings recommended in the original paper, rather than performing a grid search or manual tuning. The design choice aligns with a core motivation of adaptive optimization and metaBBO methods: to reduce the reliance on labor-intensive hyperparameter tuning. By using default settings, we ensure a fair and meaningful comparison that highlights the intrinsic advantages of adaptive strategies.

Table 4: Detailed hyperparameter configurations of baselines. $ub$ and $lb$ denote the upper and lower bounds of the search space, respectively. randn($d$) denotes sampling a $d$-dimensional vector from a standard normal distribution. All MetaBBO methods are trained on the same synthetic problem distribution as RLDEAFL Guo et al. (2025).

| Baseline | Parameter | Setting |
|---|---|---|
| **Traditional BBO methods** | | |
| RS | — | Uniform sampling within $[lb, ub]^d$ Bergstra & Bengio (2012). |
| PSO | Inertia weight $w$ | Linearly decreased from 0.9 to 0.4 over iterations Kennedy & Eberhart (1995). |
| | Coefficients $c_1/c_2$ | 2.0/2.0 Kennedy & Eberhart (1995) |
| DE | Mutation factor $F$ | 0.5 Storn & Price (1997) |
| | Crossover probability $CR$ | 0.5 Storn & Price (1997) |
| | Strategy | DE/rand/1/bin Storn & Price (1997) |
| **Adaptive optimization variants** | | |
| SAHLPSO | Adaptive parameters | Parameter ranges follow those specified in the original paper Tao et al. (2021). |
| JDE21 | Adaptive parameters | Parameter ranges follow those specified in the original paper Brest et al. (2021). |
| CMAES | Initial step size $\sigma$ | $0.3 \times (ub - lb)$ Hansen (2016) |
| | Initial mean $\mu$ | $\mu = lb + \text{randn}(d) \times (ub - lb)$ Hansen (2016) |
| **MetaBBO methods**(Training on BBOB or UAV training set) | | |
| GLEET | Training parameters | Parameter configurations are consistent with the original paper Ma et al. (2024). |
| RLDEAFL | Training parameters | Parameter configurations are consistent with the original paper Guo et al. (2025). |
| LES | Training parameters | Parameter configurations are consistent with the original paper Lange et al. (2023b). |
| GLHF | Training parameters | Parameter configurations are consistent with the original paper Li et al. (2024). |

Table 5: Hyperparameter configuration of ABOM.

| Parameter | Setting |
|---|---|
| Crossover dropout rate $p_C$ | 0.95 |
| Mutation dropout rate $p_M$ | 0.95 |
| Learning rate $\eta$ of AdamW | $1 \times 10^{-3}$ |
| Attention dimension $d_A$ | $d$ |
| MLP hidden dimension $d_M$ | $2^{\lfloor \log_2(d) \rfloor}$ |

## H    COMPARISON WITH EPOM

This section conducts a performance comparison between our method ABOM and EPOM, a recently proposed meta black-box optimization method that represents the current state of the art in zero-shot optimization Han et al. (2025). EPOM operates as a pre-trained optimization model that learns a generalizable mapping from task-specific features to optimization strategies, thereby enabling zero-shot optimization capabilities on previously unseen black-box problems. We evaluate ABOM and EPOM on the Bipedal Walker task, which requires optimizing a fully-connected neural network policy with $d = 874$ parameters over $k = 800$ timesteps to enhance robotic locomotion control performance. To ensure a fair and reproducible comparison, we strictly adhere to the experimental protocol and parameter settings

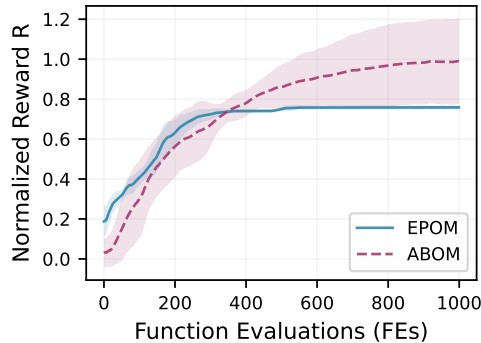

Figure 8: Convergence comparison between ABOM and EPOM on Bipedal Walker task.

established in the original paper Han et al. (2025). ABOM utilizes the identical hyperparameters as those employed in our prior experiments (refer to Table 5 for details), maintaining consistency across evaluations. As demonstrated in Fig. 8, ABOM achieves significantly faster convergence to high-quality solutions, while EPOM exhibits premature convergence, underscoring the robustness and effectiveness of our method in challenging optimization scenarios.

## I    SENSITIVITY ANALYSIS OF LEARNING RATE

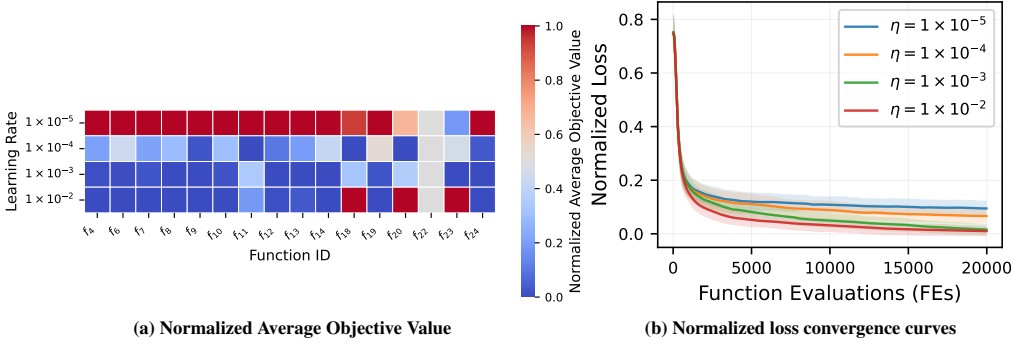

(a) Normalized Average Objective Value                    (b) Normalized loss convergence curves

Figure 9: Sensitivity analysis of learning rate $\eta$ on the BBOB suite with $d = 30$.

Fig. 9 presents the sensitivity analysis of the learning rate of AdamW for ABOM's parameter adaptation. The heatmap (Fig. 9(a)) shows optimization performance across different learning rates, while the loss curves (Fig. 9(b)) demonstrate the convergence behavior of the parameter adaptation.

The loss exhibits stable convergence across the evaluated learning rate spectrum (Fig. 9(b)), empirically validating our theoretical assumption of local convergence for parameter adaptation (Eq. 18). However, as shown in Fig. 9(a), optimization performance varies significantly across learning rates. $\eta = 1 \times 10^{-3}$ achieves the best balance between convergence speed and solution quality. This demonstrates that while convergent behavior of the loss is necessary for stable parameter adaptation, it does not guarantee optimal optimization performance. The choice of learning rate remains crucial for effective parameter adaptation.

## J    PRELIMINARY GENERALIZATION ANALYSIS OF ABOM

This section preliminarily explores the generalization capability of ABOM through pre-training on the STOP benchmark suite Xue et al. (2025). The STOP suite comprises 12 sequence transfer optimization problems, where each problem contains a series of source optimization tasks and one

target optimization task. Based on the similarity between source and target tasks (measured by fitness landscape overlap and optimal solution alignment), the 12 problems are categorized into three groups: *high similarity* (STOP1–4), *mixed similarity* (STOP5–8), and *low similarity* (STOP9–12). Detailed properties of the optimization tasks are provided in Xue et al. (2025). For experimental evaluation, each problem is instantiated with 10 source tasks under a maximum evaluation budget of 5,000 per task. We treat these source tasks as the training set and the target task as the test set. This setup spans diverse similarity scenarios between training and test sets, enabling a more comprehensive evaluation of ABOM's generalization performance.

We introduce ABOM-PT, a pre-trained variant of ABOM, where meta-optimization knowledge is distilled from the training set. Specifically, ABOM is executed on 10 source tasks for $T = 250$ iterations per task, generating 2500 prior training samples $\mathcal{M} = \{(P_k^{(t)}, F_k^{(t)}) \mid P_k^{(t)} \in \mathbb{R}^{N \times d}, F_k^{(t)} \in \mathbb{R}^N, k = 1, ..., K, t = 1, ..., T\}$. The pre-training objective minimizes the prediction error of population evolution:

$$\min_W \mathcal{L}_{\text{pt}} = \sum_{k=1}^{K} \sum_{t=1}^{T-1} \left\| P_k^{(t+1)} - \text{ABOM}_W(P_k^{(t)}, F_k^{(t)}) \right\|^2, \tag{43}$$

where $\text{ABOM}_W$ predicts the next-generation population $P_k^{(t+1)}$ from current population-fitness pairs. We optimized the Eq. (43) using AdamW with a learning rate of $1 \times 10^{-3}$ and a batch size of 256. The hyperparameters for parameter adaptation are consistent with those in Table 5.

Table 6 presents the experimental results of ABOM and ABOM-PT on the STOP benchmark suite, revealing four key insights: 1) ABOM-PT outperforms ABOM in 9 of 12 problems, confirming the generalization capability of our method; 2) Under high-similarity conditions (STOP1-4), ABOM-PT achieves substantially better performance by effectively leveraging optimization knowledge from training tasks to the test task; 3) ABOM-PT underperforms on some mixed-similarity problems (such as STOP8), revealing limitations in handling complex task relationships; 4) Surprisingly, pre-training on low-similarity tasks (STOP9-12) consistently improves performance on the test task, demonstrating that even dissimilar training tasks contain valuable optimization knowledge that enhances generalization capability.

Table 6: Performance comparison of ABOM vs. ABOM-PT on the STOP suite over 30 independent runs, reported as the mean and standard deviation of objective values (lower is better).

| Problem | Similarity | ABOM (mean ± std) | ABOM-PT (mean ± std) |
|---|---|---|---|
| STOP1 | High | 1.08e+0 ± 4.42e-1 | **4.73e-1 ± 1.97e-1** |
| STOP2 | High | 1.92e-1 ± 7.26e-2 | **2.61e-2 ± 7.45e-4** |
| STOP3 | High | 1.20e+0 ± 4.87e+0 | **1.71e-1 ± 8.04e-3** |
| STOP4 | High | 2.52e-1 ± 6.50e-3 | **2.08e-1 ± 2.72e-3** |
| STOP5 | Medium | 2.79e+0 ± 1.21e+1 | **1.28e-2 ± 9.88e-6** |
| STOP6 | Medium | 1.06e+2 ± 9.62e+2 | **1.01e+2 ± 3.74e+2** |
| STOP7 | Medium | 5.27e-2 ± 1.28e-2 | **6.84e-3 ± 7.83e-5** |
| STOP8 | Medium | **3.60e+0 ± 1.22e+1** | 5.39e+0 ± 2.90e+1 |
| STOP9 | Low | 1.75e-2 ± 1.50e-4 | **3.84e-3 ± 4.32e-6** |
| STOP10 | Low | 3.87e+1 ± 5.49e+1 | **2.69e+1 ± 4.61e+1** |
| STOP11 | Low | 5.02e+0 ± 3.97e+1 | **6.20e-1 ± 1.20e-1** |
| STOP12 | Low | 9.52e+2 ± 1.38e+6 | **8.55e+1 ± 1.20e+4** |
| $-/\approx/+$ | | 9/3/0 | - |

## K EXPERIMENTAL RESULTS

Tables 7 and 8 show the mean and standard deviation over 30 runs for each baseline on the BBOB suite with $d = 30/100$. The convergence curves of the average normalized cost across all test functions for the BBOB suite, with dimensions $d = 30/100/500$, are presented in Fig. 10, based on 30 independent runs.

The convergence curves of cost (log scale) for the 28 UAV problems over 30 independent runs are shown in Fig. 11, Fig. 12, and Fig. 13.

The boxplots of cost (log scale) over 30 independent runs for the 28 UAV problems are shown in Fig. 14, Fig. 15, and Fig. 16.

Table 7: The comparison results of the baselines on the BBOB suite with $d = 30$. All results are reported as the mean and standard deviation (mean $\pm$ std) over 30 independent runs. Symbols "$-$", "$\approx$", and "$+$" imply that the corresponding baseline is significantly worse, similar, and better than ABOM on the Wilcoxon rank-sum test with 95% confidence level, respectively. The best results are indicated in **bold**, and the suboptimal results are underlined.

| ID | Traditional BBO | | | Adaptive Variants | | | MetaBBO | | | | Ours |
|---|---|---|---|---|---|---|---|---|---|---|---|
| | RS | PSO | DE | SAHLPSO | JDE21 | CMAES | GLEET | RLDEAFL | LES | GLHF | ABOM |
| $f_4$ | 5.17e+5 ±1.25e+5 | 1.28e+5 ±3.87e+4 | 9.84e+3 ±1.80e+3 | 2.04e+5 ±2.43e+5 | 5.58e+3 ±4.39e+3 | **3.25e+1 ±6.05e+0** | 3.68e+4 ±3.44e+4 | 6.18e+3 ±3.89e+3 | 1.81e+6 ±3.30e+5 | 6.96e+5 ±4.08e+5 | 5.45e+2 ±2.95e+2 |
| $f_6$ | 5.42e+7 ±7.85e+6 | 2.49e+5 ±2.45e+5 | 6.12e+4 ±7.08e+3 | 5.11e+6 ±1.05e+7 | 2.80e+4 ±1.19e+4 | **5.33e+0 ±4.28e+0** | 3.76e+4 ±2.82e+4 | 2.09e+4 ±1.24e+4 | 8.01e+7 ±8.30e+6 | 6.61e+7 ±1.58e+7 | 2.60e+2 ±2.64e+2 |
| $f_7$ | 2.68e+5 ±4.35e+4 | 6.70e+4 ±1.85e+4 | 2.45e+4 ±4.99e+3 | 5.67e+4 ±5.23e+4 | 3.86e+3 ±1.59e+3 | 3.26e+5 ±7.09e+5 | 8.11e+3 ±5.56e+3 | 6.28e+3 ±4.18e+3 | 3.54e+5 ±3.62e+4 | 2.59e+5 ±4.79e+4 | **5.58e+2 ±2.77e+2** |
| $f_8$ | 1.45e+10 ±2.94e+9 | 1.23e+9 ±5.32e+8 | 1.27e+3 ±2.86e+3 | 2.46e+8 ±2.53e+8 | 7.85e+3 ±3.26e+4 | 5.63e+2 ±1.57e+2 | 1.29e+7 ±6.97e+7 | 1.04e+3 ±1.98e+3 | 3.82e+9 ±4.23e+8 | 4.02e+9 ±5.45e+8 | **1.15e+2 ±1.56e+2** |
| $f_9$ | 1.19e+10 ±2.13e+9 | 8.86e+8 ±2.27e+8 | 1.56e+4 ±1.53e+4 | 3.82e+7 ±3.65e+7 | 2.34e+4 ±5.40e+4 | **2.48e+1 ±1.05e+0** | 2.03e+4 ±5.28e+4 | 4.36e+4 ±1.42e+5 | 1.49e+3 ±2.12e+0 | 1.85e+2 ±1.69e+0 | 2.35e+3 ±5.30e+3 |
| $f_{10}$ | 7.47e+8 ±1.56e+8 | 1.43e+8 ±6.28e+7 | 1.11e+8 ±2.29e+7 | 2.51e+8 ±3.13e+8 | 1.73e+7 ±1.32e+7 | 8.99e+6 ±4.14e+6 | 1.54e+7 ±2.64e+7 | 7.42e+6 ±3.12e+6 | 1.81e+9 ±4.37e+8 | 6.69e+8 ±3.01e+8 | **9.72e+5 ±7.38e+5** |
| $f_{11}$ | 1.12e+5 ±1.23e+4 | 8.82e+4 ±2.84e+4 | 1.04e+5 ±1.57e+4 | 9.90e+4 ±2.55e+4 | 7.32e+4 ±2.54e+4 | 3.55e+4 ±2.96e+4 | 3.23e+4 ±3.05e+4 | 8.57e+4 ±2.88e+4 | 8.99e+5 ±7.71e+3 | 6.72e+4 ±7.71e+3 | **2.61e+4 ±1.01e+4** |
| $f_{12}$ | 1.68e+15 ±3.30e+15 | 1.38e+11 ±2.12e+11 | 1.05e+9 ±4.85e+8 | 3.46e+18 ±1.40e+19 | 1.52e+9 ±2.24e+9 | **1.03e+0 ±2.14e+0** | 3.96e+10 ±3.27e+10 | 6.45e+8 ±1.79e+9 | 8.23e+19 ±7.42e+19 | 2.56e+17 ±5.16e+17 | 5.28e+7 ±1.45e+8 |
| $f_{13}$ | 4.34e+4 ±2.36e+3 | 2.17e+4 ±2.49e+3 | 6.96e+2 ±1.15e+2 | 1.51e+4 ±5.51e+3 | 5.18e+2 ±4.89e+2 | **1.09e+0 ±1.45e+0** | 3.57e+3 ±1.81e+3 | 7.71e+2 ±1.40e+3 | 3.97e+4 ±1.40e+3 | 3.78e+4 ±2.41e+3 | 7.28e+1 ±3.07e+1 |
| $f_{14}$ | 3.52e+4 ±5.49e+3 | 5.77e+3 ±1.86e+3 | 3.76e+3 ±7.40e+2 | 5.07e+3 ±6.19e+3 | 3.72e+2 ±1.91e+2 | 3.99e+0 ±6.73e+0 | 5.69e+2 ±4.04e+2 | 1.94e+2 ±8.73e+1 | 9.99e+4 ±2.17e+4 | 3.30e+4 ±1.75e+4 | **3.46e-2 ±3.39e-2** |
| $f_{18}$ | 1.42e+5 ±1.06e+5 | 9.43e+2 ±2.09e+2 | 6.03e+2 ±6.10e+1 | 8.89e+5 ±1.19e+6 | 5.61e+2 ±1.26e+2 | 3.36e+12 ±2.84e+11 | 5.79e+2 ±1.62e+2 | 5.21e+2 ±1.28e+2 | 2.48e+6 ±1.74e+6 | 5.18e+5 ±4.19e+5 | **5.12e+2 ±1.37e+2** |
| $f_{19}$ | 1.01e+6 ±1.89e+5 | 7.82e+4 ±1.98e+4 | 1.22e+1 ±3.78e+0 | 2.92e+3 ±2.84e+3 | 2.01e+1 ±1.97e+1 | 6.76e+0 ±7.40e-1 | 1.37e+1 ±8.35e+0 | 2.87e+1 ±3.03e+1 | 1.30e+3 ±1.27e+0 | 9.19e+0 ±5.91e+0 | **2.48e-1 ±1.11e-3** |
| $f_{20}$ | 1.24e+5 ±2.78e+6 | 9.79e+5 ±6.89e+5 | -3.88e+1 ±2.75e+0 | 4.04e+3 ±1.62e+4 | -6.29e+1 ±2.67e+0 | -1.53e+1 ±9.73e+0 | -3.97e+1 ±4.98e+0 | -5.20e+1 ±5.06e+0 | 1.49e+3 ±2.37e+0 | -4.10e+0 ±1.77e+0 | **-6.57e+1 ±3.80e+0** |
| $f_{22}$ | 8.66e+1 ±0.00e+0 | 8.66e+1 ±0.00e+0 | 8.66e+1 ±0.00e+0 | 8.66e+1 ±0.00e+0 | 8.66e+1 ±0.00e+0 | 8.66e+1 ±0.00e+0 | 8.66e+1 ±0.00e+0 | 8.66e+1 ±0.00e+0 | 1.19e+3 ±4.55e-13 | 8.66e+1 ±0.00e+0 | **8.66e+1 ±0.00e+0** |
| $f_{23}$ | 3.29e+0 ±3.96e-1 | 3.23e+0 ±3.82e-1 | 3.18e+0 ±4.56e-1 | 3.41e+0 ±4.94e-1 | 3.32e+0 ±3.92e-1 | 3.30e+0 ±3.68e-1 | 2.98e+0 ±3.34e-1 | 3.31e+0 ±5.37e-1 | 1.30e+3 ±3.11e-1 | 3.34e+0 ±3.70e-1 | **3.01e-1 ±2.19e-1** |
| $f_{24}$ | 1.46e+5 ±1.37e+4 | 3.84e+4 ±7.79e+3 | 3.23e+2 ±5.65e+1 | 1.22e+4 ±5.97e+3 | 5.70e+2 ±8.02e+2 | 2.58e+2 ±1.64e+1 | 3.98e+2 ±7.61e+1 | 7.08e+2 ±5.00e+2 | 5.35e+4 ±2.34e+3 | 6.11e+4 ±2.70e+3 | **2.44e+2 ±2.37e+1** |
| $-/\approx/+$ | 15/1/0 | 15/1/0 | 15/1/0 | 15/1/0 | 15/1/0 | 10/1/5 | 15/1/0 | 15/1/0 | 15/0/1 | 14/1/1 | - |

Table 8: The comparison results of the baselines on the BBOB suite with $d = 100$. All results are reported as the mean and standard deviation (mean ± std) over 30 independent runs. Symbols "−", "≈", and "+" imply that the corresponding baseline is significantly worse, similar, and better than ABOM on the Wilcoxon rank-sum test with 95% confidence level, respectively. The best results are indicated in **bold**, and the suboptimal results are underlined.

| ID | Traditional BBO | | | Adaptive Variants | | | MetaBBO | | | | Ours |
|---|---|---|---|---|---|---|---|---|---|---|---|
| | RS | PSO | DE | SAHLPSO | JDE21 | CMAES | GLEET | RLDEAFL | LES | GLHF | ABOM |
| $f_4$ | 9.64e+6 ±1.21e+6 | 1.83e+6 ±4.72e+5 | 7.23e+5 ±7.94e+4 | 2.49e+6 ±2.14e+6 | 2.82e+5 ±5.83e+4 | **3.72e+3 ±9.84e+2** | 3.77e+5 ±1.74e+5 | 2.89e+5 ±8.20e+4 | 1.14e+7 ±8.00e+5 | 9.66e+6 ±1.47e+6 | 9.72e+3 ±5.28e+3 |
| $f_6$ | 6.11e+8 ±3.81e+7 | 4.76e+7 ±2.94e+7 | 1.03e+6 ±4.92e+5 | 4.50e+7 ±3.58e+7 | 7.16e+5 ±1.07e+6 | **2.15e+4 ±4.87e+3** | 4.27e+5 ±9.26e+5 | 1.96e+5 ±9.09e+4 | 5.71e+8 ±1.63e+7 | 5.45e+8 ±2.78e+7 | 7.92e+4 ±3.49e+4 |
| $f_7$ | 1.75e+6 ±9.77e+4 | 6.23e+5 ±6.59e+4 | 4.94e+5 ±3.66e+4 | 4.76e+5 ±2.10e+5 | 9.32e+4 ±2.85e+4 | 8.81e+3 ±2.97e+3 | 7.92e+4 ±2.07e+4 | 1.17e+5 ±2.87e+4 | 1.13e+6 ±6.30e+4 | 1.08e+6 ±8.55e+4 | **6.23e+3 ±1.50e+3** |
| $f_8$ | 4.33e+11 ±3.31e+10 | 8.71e+10 ±1.59e+10 | 2.67e+8 ±5.05e+7 | 2.65e+10 ±1.53e+10 | 4.60e+9 ±4.18e+9 | 5.16e+3 ±5.73e+3 | 1.41e+8 ±1.39e+8 | 3.79e+9 ±3.66e+9 | 5.24e+10 ±2.10e+9 | 5.30e+10 ±2.68e+9 | **3.84e+3 ±5.03e+3** |
| $f_9$ | 2.87e+11 ±2.82e+10 | 7.21e+10 ±1.21e+10 | 1.53e+9 ±4.45e+8 | 1.34e+9 ±5.55e+8 | 4.97e+8 ±4.42e+8 | 5.83e+4 ±1.89e+4 | 6.37e+7 ±2.21e+7 | 3.41e+8 ±2.71e+8 | 2.96e+3 ±2.16e+1 | **6.43e+2 ±4.10e-1** | 1.13e+5 ±2.79e+5 |
| $f_{10}$ | 7.88e+9 ±7.69e+8 | 1.97e+9 ±3.14e+8 | 2.30e+9 ±2.58e+8 | 1.88e+9 ±9.32e+8 | 4.37e+8 ±1.37e+8 | 1.20e+8 ±3.66e+7 | 2.55e+8 ±8.80e+7 | 2.07e+8 ±8.93e+7 | 6.53e+9 ±5.19e+8 | 5.63e+9 ±8.08e+8 | **6.65e+7 ±1.48e+7** |
| $f_{11}$ | 4.30e+5 ±2.62e+4 | 4.12e+5 ±5.51e+4 | 4.35e+5 ±3.31e+4 | 2.99e+5 ±5.28e+4 | 3.78e+5 ±4.03e+4 | 9.10e+5 ±5.31e+4 | **1.97e+5 ±3.31e+4** | 2.02e+5 ±4.91e+4 | 1.15e+6 ±1.40e+6 | 2.07e+5 ±8.84e+3 | 3.48e+5 ±6.24e+4 |
| $f_{12}$ | 1.84e+22 ±2.12e+22 | 7.09e+15 ±3.37e+16 | 6.78e+13 ±6.93e+13 | 2.77e+22 ±9.39e+22 | 5.29e+12 ±8.04e+12 | **8.76e+8 ±5.66e+8** | 2.84e+11 ±1.10e+11 | 4.77e+11 ±6.81e+11 | 6.99e+22 ±5.86e+22 | 3.75e+21 ±9.78e+21 | 3.97e+9 ±1.31e+10 |
| $f_{13}$ | 1.08e+5 ±2.50e+3 | 6.70e+4 ±4.61e+3 | 2.86e+4 ±1.90e+3 | 5.28e+4 ±1.10e+4 | 2.32e+4 ±6.09e+3 | **1.09e+2 ±2.35e+1** | 1.69e+4 ±4.78e+3 | 2.31e+4 ±4.78e+3 | 8.37e+4 ±1.15e+3 | 8.14e+4 ±1.57e+3 | 3.65e+2 ±1.14e+2 |
| $f_{14}$ | 3.10e+5 ±3.56e+4 | 4.73e+4 ±9.85e+3 | 6.79e+4 ±6.41e+3 | 5.84e+4 ±5.22e+4 | 8.14e+3 ±2.57e+3 | 1.53e+3 ±3.75e+2 | 4.62e+3 ±1.97e+3 | 4.89e+3 ±1.58e+3 | 2.60e+5 ±2.62e+4 | 2.16e+5 ±3.70e+4 | **3.33e+2 ±1.35e+2** |
| $f_{18}$ | 5.81e+7 ±2.84e+7 | 6.42e+3 ±8.93e+3 | 1.59e+4 ±8.45e+3 | 7.27e+7 ±3.12e+8 | 1.04e+4 ±6.34e+3 | 2.82e+3 ±3.60e+3 | 4.63e+3 ±1.80e+3 | 2.68e+4 ±6.79e+3 | 2.49e+7 ±1.01e+7 | 1.12e+7 ±6.30e+6 | **2.46e+2 ±4.41e+1** |
| $f_{19}$ | 7.48e+6 ±5.96e+5 | 1.77e+6 ±2.80e+5 | 3.42e+4 ±7.96e+3 | 2.87e+4 ±1.42e+4 | 1.06e+4 ±8.60e+3 | 2.93e+1 ±2.54e+0 | 1.69e+3 ±7.41e+2 | 6.34e+3 ±6.13e+3 | 5.06e+2 ±1.65e+0 | 1.54e+1 ±1.16e+1 | **2.50e-1 ±7.63e-5** |
| $f_{20}$ | 1.16e+8 ±5.88e+6 | 3.44e+7 ±6.40e+6 | 3.05e+4 ±3.04e+4 | 2.33e+5 ±2.88e+5 | 7.12e+4 ±1.44e+5 | -3.04e+0 ±1.72e+0 | -1.92e+1 ±5.28e+0 | 7.93e+4 ±1.51e+5 | 2.10e+3 ±8.78e-1 | 9.08e-1 ±5.31e-1 | **-5.60e+1 ±3.36e+0** |
| $f_{22}$ | 8.66e+1 ±0.00e+0 | 8.66e+1 ±0.00e+0 | 8.66e+1 ±0.00e+0 | 8.66e+1 ±0.00e+0 | 8.66e+1 ±0.00e+0 | 8.66e+1 ±0.00e+0 | 8.66e+1 ±0.00e+0 | 8.66e+1 ±0.00e+0 | 1.29e+3 ±4.55e-13 | 8.66e+1 ±0.00e+0 | **8.66e+1 ±0.00e+0** |
| $f_{23}$ | 4.85e+0 ±3.86e-1 | 4.87e+0 ±2.39e-1 | 4.90e+0 ±3.87e-1 | 4.80e+0 ±4.32e-1 | 4.85e+0 ±2.27e-1 | 7.11e+0 ±6.07e-1 | 4.84e+0 ±3.44e-1 | 4.66e+0 ±6.00e-1 | 2.11e+3 ±2.98e-1 | 4.88e+0 ±3.51e-1 | **4.61e+0 ±4.11e-1** |
| $f_{24}$ | 9.01e+5 ±3.01e+4 | 4.31e+5 ±3.66e+4 | 2.03e+4 ±1.26e+4 | 1.58e+5 ±5.12e+4 | 3.36e+4 ±1.53e+4 | 1.21e+3 ±7.02e+1 | 9.89e+3 ±1.84e+3 | 3.66e+4 ±1.64e+4 | 2.26e+5 ±3.33e+3 | 2.35e+5 ±3.69e+3 | **1.05e+3 ±3.56e+1** |
| −/≈/+ | 14/2/0 | 14/2/0 | 14/2/0 | 13/2/1 | 14/2/0 | 9/2/5 | 13/2/1 | 13/2/1 | 15/0/1 | 12/2/2 | - |

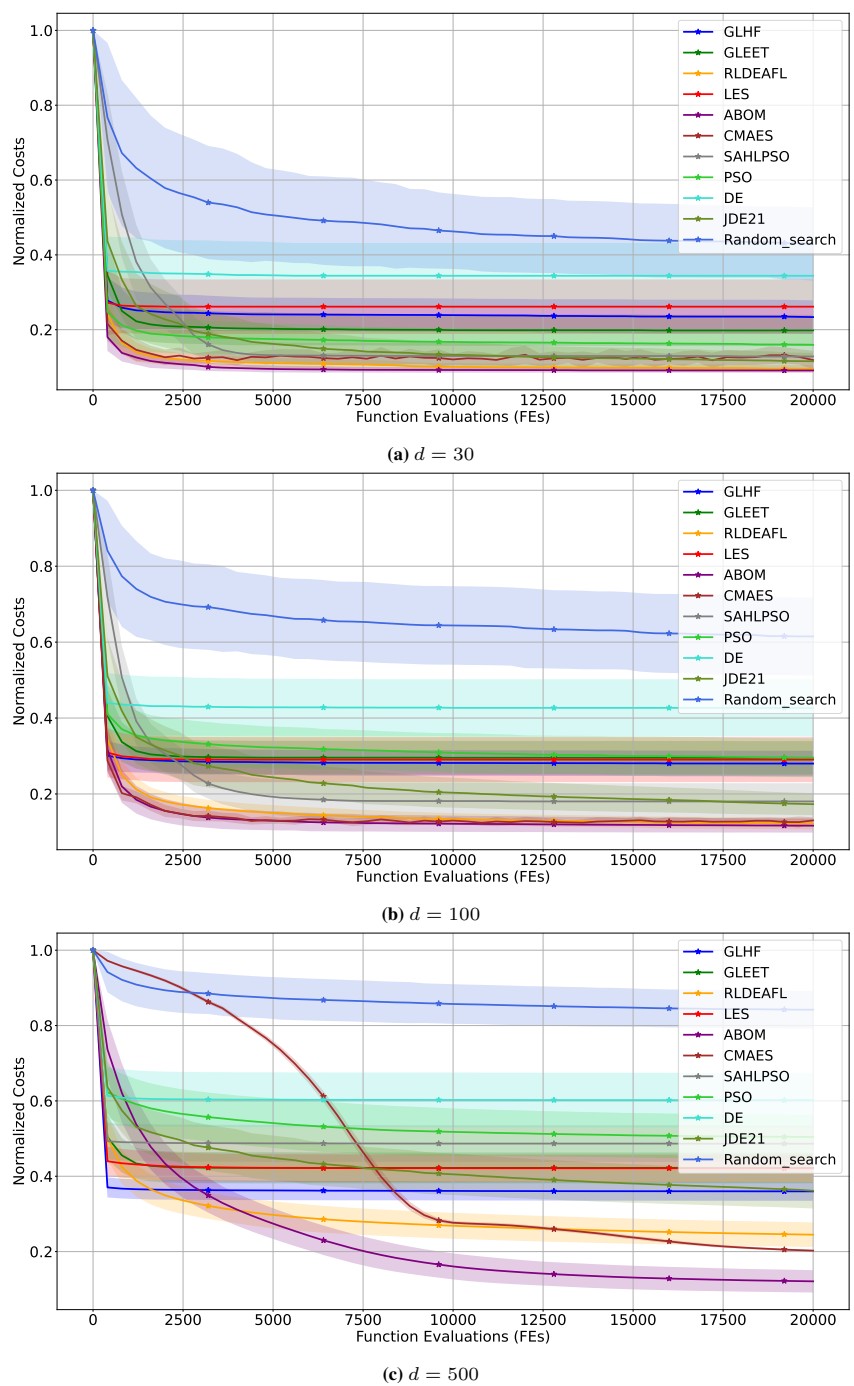

**(a)** $d = 30$

**(b)** $d = 100$

**(c)** $d = 500$

Figure 10: Convergence curves of the average normalized cost across all test functions in the BBOB suite, with $d = 30/100/500$, over 30 independent runs. The cost values are min-max normalized per function to ensure comparability. Each subplot displays the performance trend, highlighting the algorithm's scalability as the dimensionality increases.

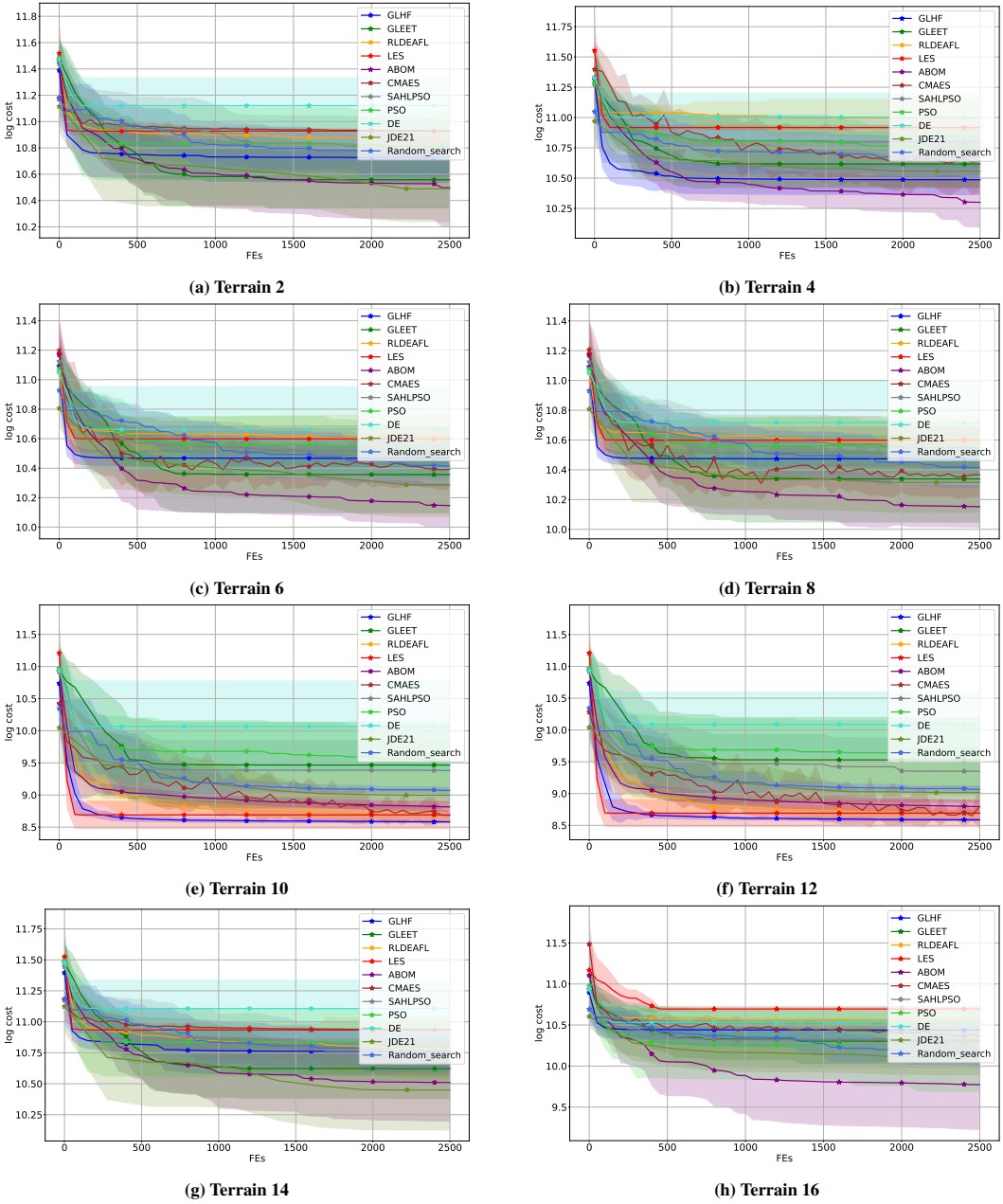

Figure 11: Convergence curves of cost (log scale) for UAV problems (Terrain 2 to 16).

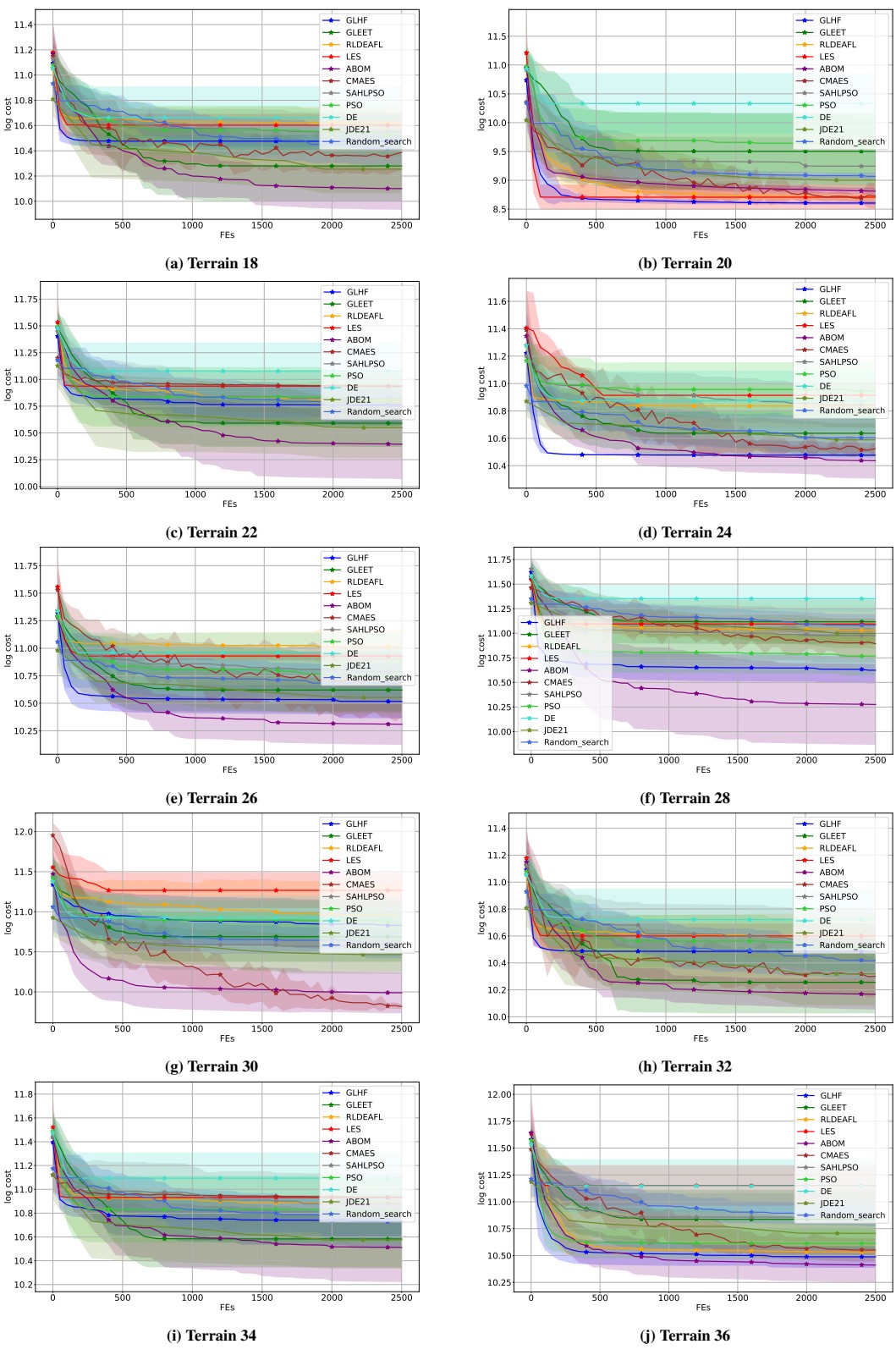

Figure 12: Convergence curves of cost (log scale) for UAV problems (Terrain 18 to 36).

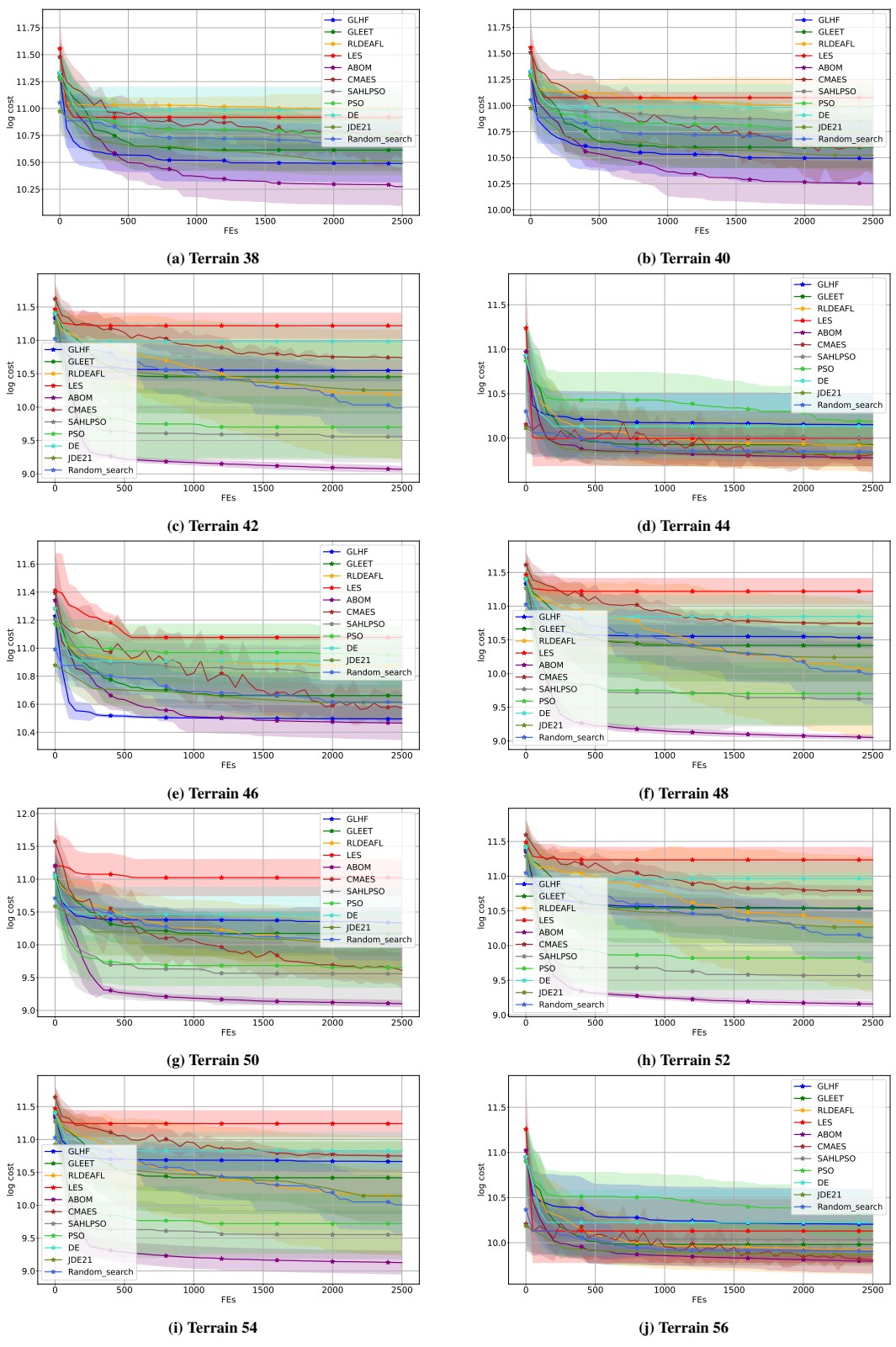

Figure 13: Convergence curves of cost (log scale) for UAV problems (Terrain 38 to 56).

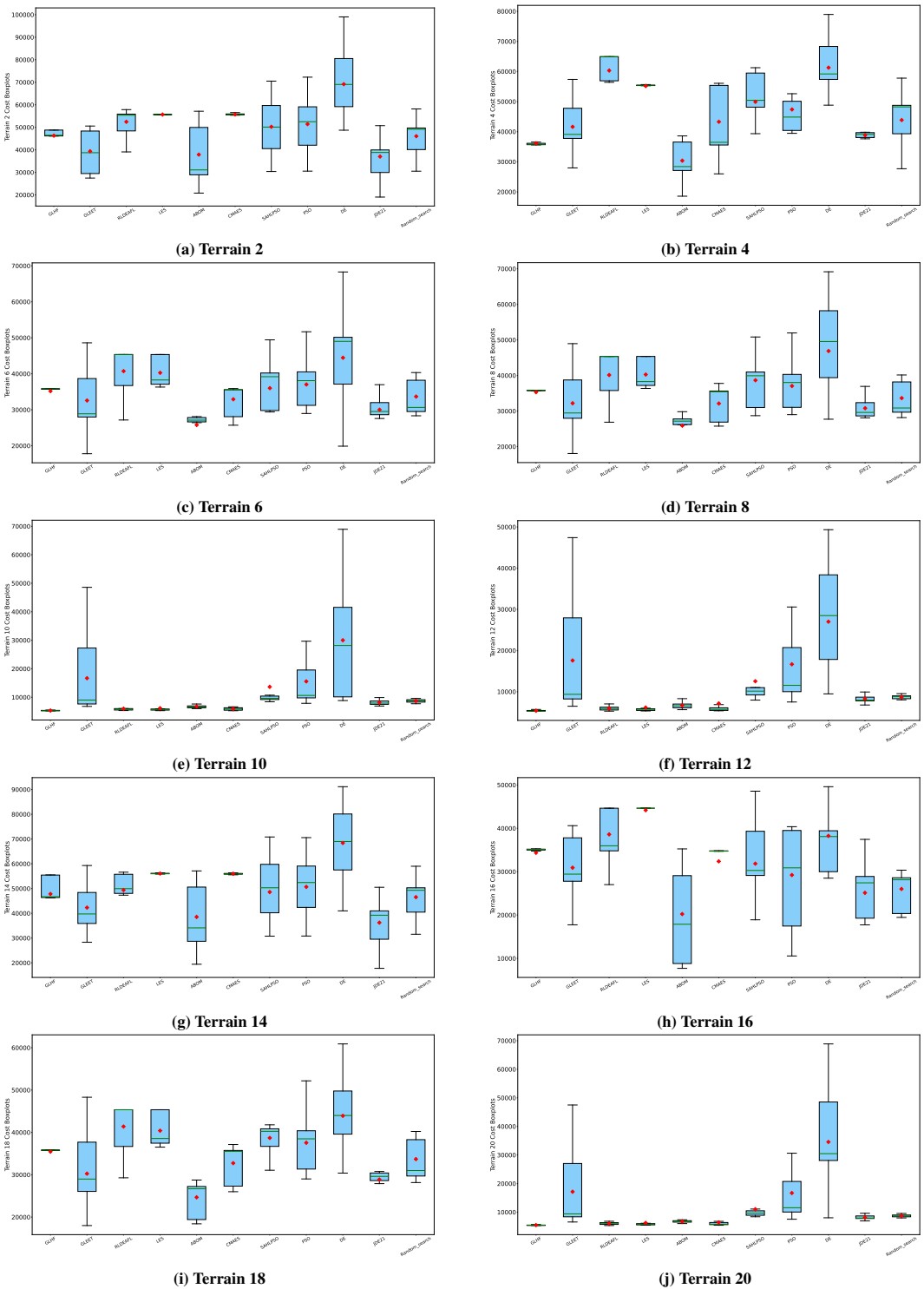

Figure 14: Boxplots of cost (log scale) over 30 runs for UAV problems (Terrain 2 to 20).

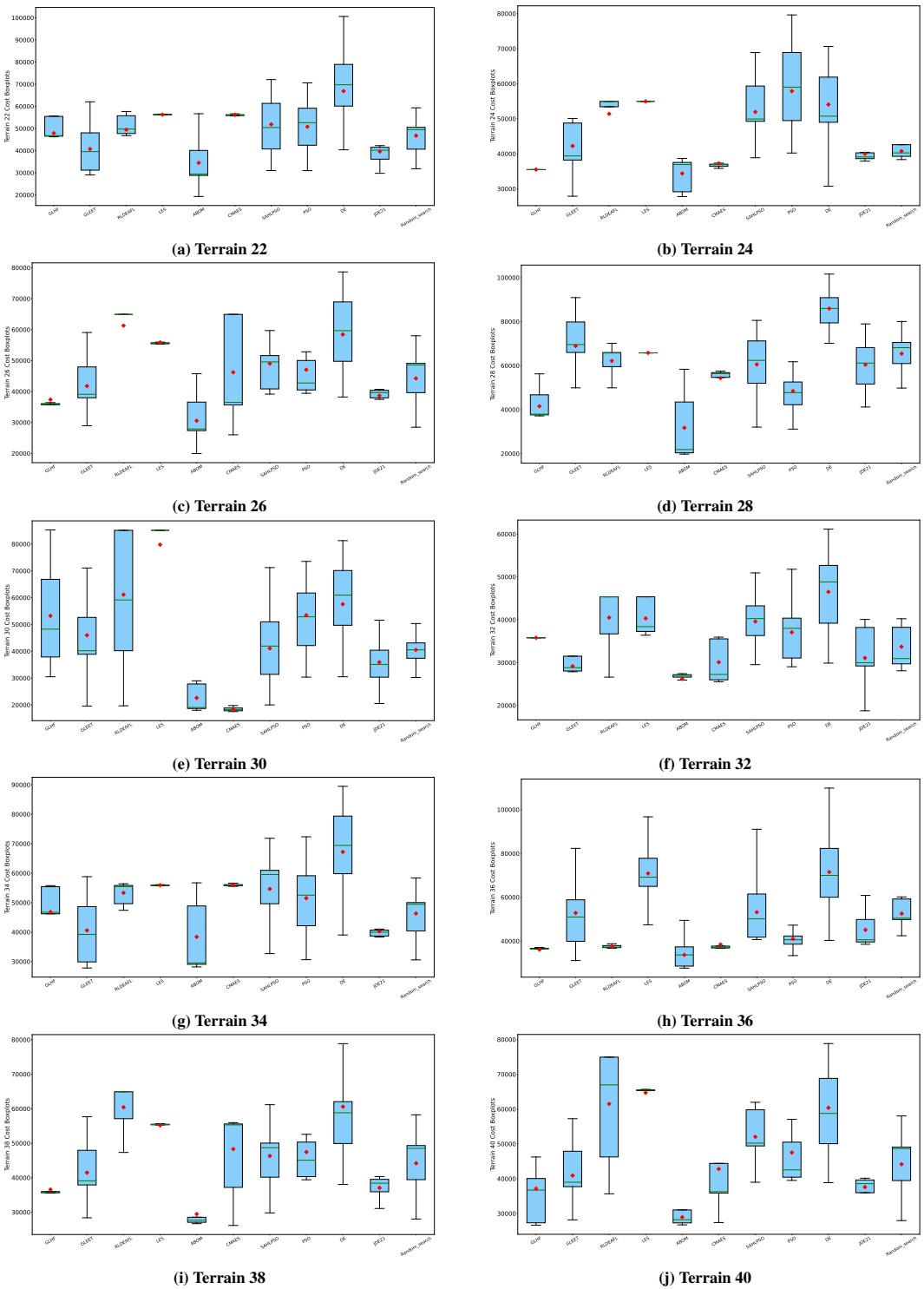

Figure 15: Boxplots of cost (log scale) over 30 runs for UAV problems (Terrain 22 to 40).

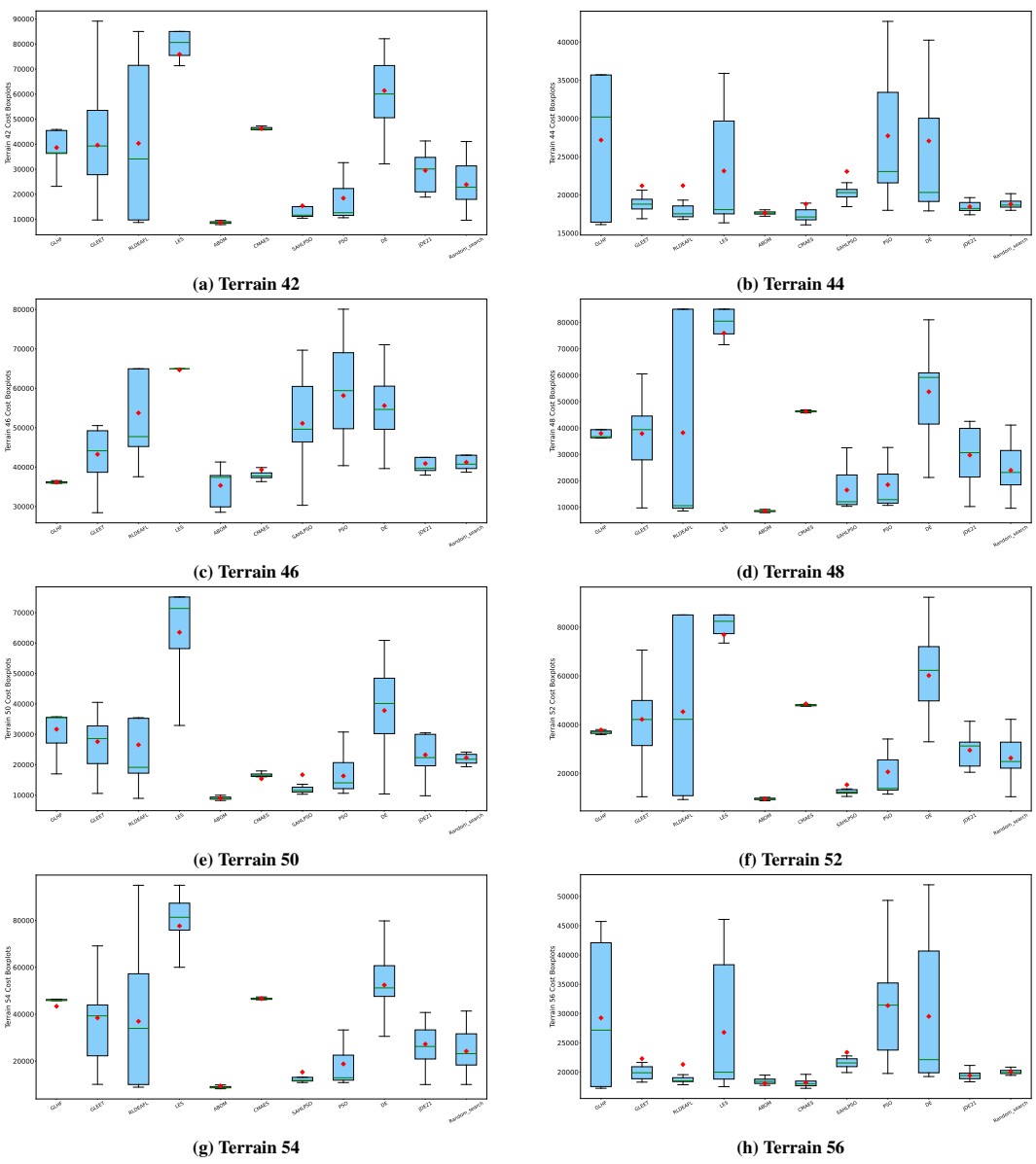

Figure 16: Boxplots of cost (log scale) over 30 runs for UAV problems (Terrain 42 to 56).

