# OpenReview forum: "Task-free Adaptive Meta Black-box Optimization"
_ICLR.cc/2026/Conference — ICLR 2026 Oral_

### Official Review · Reviewer_tkpd · 2025-10-28

**Soundness:** 2
**Presentation:** 2
**Contribution:** 2
**Rating:** 2
**Confidence:** 3

**Summary:**

This paper proposes ABOM, a black-box optimization algorithm which adaptively adjust parameters based on experience from function evaluations. ABOM parameterizes the selection, crossover and mutation module in evolutionary algorithm via MLP with attention mechanism, and train these parameters by approximating to the elite selection procedure. The authors provide the global convergence guarantee of ABOM. The experimental results over both synthetic and UAV benchmarks demonstrate that ABOM matches or exceeds commonly used BBO baselines.

**Strengths:**

1. The proposed MLP parameterized algorithm can benefit from GPU parallel computation.

2. ABOM demonstrates superior empirical optimization performances on both synthetic and UAV benchmarks.

**Weaknesses:**

1. The theoretical and empirical algorithm analysis does not explain the reason of using attention-based MLP to parameterize the evolutionary modules. From the algorithm design side, the "optimal" policy corresponds to a elite-selection style EA algorithm. In the theoretical analysis, the main results are derived by setting all parameters except the last layer's bias to 0. The global convergence guarantee is not finite-time, which seems to hold simply because of the introduced randomness from dropout. I think other EA based on elite selection might also fit to this infinite-time convergence guarantee.

2. The proposed ABOM also introduces additional hyperparameters such as dropout rate and hidden dimensions, which is sensitive according to figure 5 and non-trivial to tune like other BBO algorithms such as CMA-ES.

3. The target problem of the proposed method should be single-task black-box optimization problem, where algorithm sample strategies and parameters are usually adjusted based on the sampled data (e.g., MLE-based parameter fitting in Bayesian optimization, and step-size adjustment in CMA-ES). Therefore, I think meta BBO problem and methods are not much related to the main research topic of this paper.

4. I think adding more recent EA-based or NN-based BBO baselines such as [1, 2] and more challenging high-dimensional benchmarks used in these works can better assess the algorithm performance.

5. (minor) The presentation can be improved to be clearer. (1) The citation format should be \citep or \citet instead of \cite to compatible with ICLR templates. (2) The population size and elite number both use notation $N$.

[1] Zhang, Zhengfei, Yunyue Wei, and Yanan Sui. "An invariant information geometric method for high-dimensional online optimization." 6th Annual Learning for Dynamics & Control Conference. PMLR, 2024.

[2] Yun, Taeyoung, et al. "Posterior Inference with Diffusion Models for High-dimensional Black-box Optimization." Forty-second International Conference on Machine Learning.

**Questions:**

1. According to Algorithm 1, is there only one gradient update in each sampling iteration?

2. How does ABOM handle the boundary issue, since MLP parameterized policy may generate samples outside the input bound.

---

> ### Author Response · Authors · 2025-11-21
> **Response to Weaknesses 1–3**
>
> We thank the reviewer for the insightful comments. We have carefully revised the manuscript to address these concerns, with all modifications highlighted in red in the revised version.
>
> **1. Response to Weakness 1 (Algorithm Design and Theoretical Analysis)**
> We appreciate your insightful observation about the theoretical analysis and clarify our contributions:
>
> - **Attention-based MLP serves three critical purposes**: 1) **Learnability**: Attention mechanisms model relationships among individuals, fitness landscapes, and genetic components, enabling gradient-based adaptation of evolutionary operators; 2) **GPU acceleration**: Neural computation enables efficient batched execution on GPU (**Fig. 3b** shows 3.5× speedup over CPU); 3) **Interpretability**: Attention matrices reveal structured search patterns (**Fig. 4**), providing insights into selection bias and genetic interactions.
>
> - **ABOM learns locally optimal operators**, not a global "optimal policy". In each generation, the current population and elite archive constitute a sample from which we derive task-specific operator parameters through a single-step gradient update with AdamW. This **local adaptation mechanism** enables ABOM to dynamically adjust its search behavior according to the evolving fitness landscape, as empirically validated by **Fig. 4**, which demonstrates distinct update rules learned at different optimization stages.
>
> - **Theoretical analysis with simplified parameters** establishes that global convergence fundamentally stems from the elite preservation mechanism and dropout-enabled exploration, rather than from the specific parameterization. **Fig. 7** shows both ABOM and ABOM-NPA converge, confirming that **convergence stems from our core mechanisms**. This does not imply "optimal" policy corresponds to zero attention weights; rather, it establishes a theoretical foundation showing convergence is maintained regardless of parameter adaptation.
>
> - **Our theoretical contribution distinguishes ABOM from other MetaBBO methods**: We prove that the attention-based MLP parameterization with dropout maintains persistent exploration capability (**Lemma 1**), and global convergence is preserved despite adaptive parameter learning (**Theorem D.1**). This addresses a critical gap in methods like GLHF and B2Opt, which lack rigorous convergence analysis despite empirical success.
>
> **2. Response to Weakness 2 (Hyperparameter Sensitivity)**
> We thank the reviewer for this important question:
>
> - Our experiments in **Fig. 5** confirm that dropout rate and hidden dimensions exhibit sensitivity to different problem characteristics. However, **ABOM demonstrates strong parameter robustness in practice**. All experiments across both BBOB and UAV benchmarks use identical parameter settings (**Table 5**) while achieving state-of-the-art results.
>
> - **This consistent performance across diverse problems demonstrates ABOM's ability to maintain effectiveness despite parameter sensitivity**. Dynamically adapting hyperparameters represents a valuable research direction, which we have added to **future work** as a promising extension that could further enhance optimization performance.
>
> **3. Response to Weakness 3 (Single-task BBO vs. MetaBBO)**
> We agree ABOM solves single-task black-box optimization problems but belongs to MetaBBO because it implements **"on-instance meta-learning"** within the optimization process:
>
> - ABOM falls under MetaBBO's **"solution manipulation" category [1]**, where meta-strategies are integrated directly into the optimization process. At each generation, ABOM treats the current population as a task and adapts operator parameters via single-step gradient updates. This creates an **inner optimization loop that learns how to optimize within the outer optimization loop**.
>
> - **Fig. 6** validates the convergence of this on-instance meta-learning method by showing that ABOM's parameter adaptation converges consistently across high-dimensional BBOB functions. Unlike methods like CMA-ES that use hand-designed update rules, ABOM **learns optimization strategies through self-supervised learning**.
>
> - This perspective doesn't conflict with single-task BBO; rather, it represents an **evolution of MetaBBO that performs meta-learning within single tasks** rather than across tasks.
>
> [1] Toward automated algorithm design: A survey and practical guide to meta-black-box-optimization. IEEE TEVC, 2025.

---

> ### Author Response · Authors · 2025-11-21
> **Response to Weaknesses 4-5 and Questions 1-2**
>
> **4. Response to Weakness 4 (Baseline Methods and Benchmarks)**
> We appreciate your suggestion to enhance our experimental evaluation:
>
> - We have compared ABOM with **EPOM [1]**, which represents the state-of-the-art in MetaBBO. While methods like Zhang (2024) and Yun (2024) are interesting, they are not directly comparable as they **are not based on evolutionary meta black-box optimization**.  And their motivations are unrelated to the field of meta black-box optimization.
>
> - We added a **neuro-evolution task (Bipedal Walker, d = 874)** to evaluate ABOM and EPOM. **Fig. 8 (Appendix H)** shows that ABOM achieves **superior performance compared to EPOM** on this challenging high-dimensional benchmark.
>
> [1] Enhancing zero-shot black-box optimization via pretrained models with efficient population modeling, interaction, and stable gradient approximation. In NeurIPS, 2025.
>
> **5. Response to Weakness 5 (Presentation Improvements)**
> We appreciate your suggestions for improving presentation clarity:
>
> - **All citations have been updated** to \citet format for consistency.
>
> - **Notation clarification**: We use the same symbol $N$ for both population size and elite number because ABOM follows the classic Elitism principle (as in NSGA-II [1]), where the elite archive size equals population size. This design choice is **essential for our loss function**, which requires alignment between offspring and elite archive.
>
> [1] A fast and elitist multiobjective genetic algorithm: Nsga-ii. IEEE TEVC, 2002.
>
> **6. Response to Question 1 (Gradient Update Frequency)**
> Algorithm 1 performs exactly **one gradient update per sampling iteration**:
>
> - The generated offspring and elite archive at each generation constitute **a single sample for parameter adaptation** ($\min_{\boldsymbol{\theta}} \mathcal{L}^{(t)} = \| \hat{\mathbf{P}}^{(t)} - \mathbf{E}^{(t)} \|^2$), requiring only one AdamW update. This aligns with our theoretical assumption of local convergence for parameter adaptation (Eq. 18).
>
> - **Fig. 4** demonstrates that ABOM learns distinct update rules at different optimization stages, while **Fig. 7 (Appendix E)** empirically validates that the parameter adaptation process **consistently converges across the BBOB suite**.
>
> **7. Response to Question 2 (Boundary Handling)**
> ABOM handles boundary issues as follows:
>
> - **Known boundaries** (e.g., BBOB problems): When offspring exceed search boundaries, we clip them to the nearest boundary value, following standard MetaBBO practice as implemented in the MetaBox platform [1].
>
> - **Unknown boundaries** (e.g., UAV path planning and neuro-evolution): We do not perform explicit boundary handling. Remarkably, ABOM still achieves **excellent performance (Figs. 3 and 8)**, demonstrating its **robustness to boundary conditions**.
>
> - **Theoretical consideration**: Our convergence analysis (Assumption 2) assumes the global optimum lies in the interior of the search space, which is common in evolutionary algorithm theory [2].
>
> [1] Metabox-v2: A unified benchmark platform for meta-black-box optimization. NeurIPS, 2025.
>
> [2] Evolutionary learning: Advances in theories and algorithms. Springer, 2019.
>
> We sincerely appreciate your thoughtful feedback, which has helped us significantly improve the clarity and rigor of our paper.

---

> > ### Comment · Reviewer_tkpd · 2025-11-21
> >
> > I thank the authors for their detailed feedback. I have some follow-up questions.
> >
> > **Regarding W1:**
> > -  "global convergence fundamentally stems from the elite preservation mechanism and dropout-enabled exploration." Does this imply that any elite-based stochastic algorithm, such as vanilla ES, would inherit the same infinite-time global convergence property?
> >
> > - In Fig. 7, ABOM and ABOM-NPA converge to different values despite both having global convergence guarantees. Could the authors clarify the underlying cause of this discrepancy?
> >
> > **Regarding W3 & W4:**
> > - "it represents an Evolution of MetaBBO that performs meta-learning within single tasks." I think this problem setting appears fundamentally different from standard MetaBBO formulations in which algorithms are pretrained on a distribution of tasks and then deployed on unseen tasks. Therefore, I think comparing ABOM with MetaBBO methods that rely on multi-task pretraining and keep parameters fixed at deployment may not be fully appropriate.
> >
> > - Since the outer optimization loop in ABOM remains a single-task BBO problem, ABOM differs from traditional single-task BBO methods mainly in its sampling strategy and parameter-update procedure. Therefore, I believe it is necessary to compare ABOM against state-of-the-art single-task BBO algorithms, rather than restricting the baselines to EA-based MetaBBO methods.

---

> > > ### Author Response · Authors · 2025-11-21
> > > **Regarding W1 (Convergence Analysis Clarification)**
> > >
> > > We sincerely thank the reviewer for these insightful questions that help us clarify ABOM's theoretical and methodological contributions. Below we address each point with concrete evidence:
> > >
> > > **Q1: Does elite-based stochastic algorithms inherently have infinite-time convergence?**
> > >
> > > We fully agree that **elite preservation provides a *necessary condition* for convergence** in evolutionary algorithms. However, **ABOM's theoretical contribution is distinct**:
> > >
> > > - **Lemma 1** proves that **dropout-enabled exploration maintains persistent coverage of the search space** even during parameter adaptation, which shows that *MLP-based parameterization with Dropout* maintains persistent exploration capability—a property **often assumed** (eg: Mutation Module in B2Opt AAAI2025) but **rarely proven**.
> > >
> > > - **Theorem D.1** establishes that ***the adaptive parameter learning process itself* does not compromise asymptotic convergence**—unlike other MetaBBO methods, where parameter learning **lacks convergence guarantees**. This is particularly significant because:
> > >   **Asymptotic convergence forms the theoretical foundation for finite-time analysis** (e.g., drift analysis for first hitting time). In addition, drift analysis typically depends on problem-specific properties [1]. **This paves the way for future finite-time analysis** of adaptive operators on specific problems—a direction we've added to Section 5 (Future Work).
> > >
> > > **Crucially**, this doesn't mean "any elite-based algorithm has the same guarantee": it means **ABOM is the *first* MetaBBO method that *combines adaptive operator learning with provable exploration and convergence***.
> > >
> > > **Q2: Why do ABOM and ABOM-NPA converge to different values?**
> > >
> > > This reflects ***finite-time performance differences under practical evaluation budgets***, not a theoretical contradiction:
> > >
> > > **Theoretical guarantee**: **ABOM converges to the global optimum *asymptotically*** (Theorem D.1), but **finite-time behavior depends critically on how well the algorithm adapts to landscape properties**—a key challenge in evolutionary computation where rigorous first-hitting-time proofs require tailored assumptions [1].
> > >
> > > **Mechanism difference**: **ABOM-NPA uses *fixed operator parameters* throughout optimization**, resulting in a **static search strategy that cannot adapt to the current fitness landscape** (e.g., constant mutation rule fails to adjust when trapped in local optima for **multi-modal** functions). In contrast, **ABOM learns *dynamic update rules* through attention-based operators**. As shown in Fig. 4, our parameterized operators **adapt selection and mutation criteria across evolutionary stages**. For example, we observe a clear progression of the select matrix: **early generations show diverse selection** (exploration phase), **middle generations focus on promising regions** (balancing phase), and **later generations concentrate on high-quality solutions** (exploitation phase).
> > >
> > > **Practical significance**: Within **limited evaluations** (e.g., 20k function calls), this adaptability allows ABOM to reach ***superior solutions*** (Fig. 7), directly demonstrating **the effectiveness of our parameter learning** while maintaining the same asymptotic guarantee. **The performance gap validates that adaptive operators outperform fixed ones in realistic scenarios**.
> > >
> > > [1 ]Evolutionary learning: Advances in theories and algorithms. Springer, 2019.

---

> ### Author Response · Authors · 2025-11-21
> **Regarding W3 & W4 (MetaBBO Formulation and Baseline Comparison)**
>
> **1. ABOM's Alignment with MetaBBO Framework**
>
> **We appreciate the reviewer's insightful comments regarding the MetaBBO formulation.** We would like to clarify how ABOM aligns with the established understanding of MetaBBO in the literature. As recognized in recent surveys (Li et al., IEEE TEVC 2025), MetaBBO is fundamentally defined by its **bilevel optimization structure**, where the upper-level (meta) configures optimization strategies $J(\boldsymbol{\theta})$ and the lower-level applies these strategies to solve black-box problems $\min_{\mathbf{x} \in \mathbb{R}^d} f_T(\mathbf{x})$. ABOM precisely follows this structure: the upper-level learns parameters of optimization strategies ($\min_{\boldsymbol{\theta}} \mathcal{L}^{(t)} = \| \hat{\mathbf{P}}^{(t)} - \mathbf{E}^{(t)} \|^2$), while the lower-level uses these operators to solve the optimization problem. This is a typical example of **"algorithm configuration" and "solution manipulation"** in MetaBBO.
>
> **2. ABOM's Extension to the MetaBBO Framework**
>
> **Our approach extends the MetaBBO framework** by utilizing *in-task optimization data* as a training signal. While traditional MetaBBO methods typically train on optimization data from prior task distributions, ABOM leverages optimization data generated from the current task itself. This extension remains fully consistent with MetaBBO's mathematical definition. **Notably, ABOM represents a meaningful expansion of MetaBBO capabilities rather than a replacement**—ABOM's task-free adaptation and prior knowledge integration are **complementary capabilities, not competing approaches**. This represents a promising research direction: a hybrid training paradigm would allow ABOM to function both as a **task-free optimizer for unknown distributions** and as a **knowledge-enhanced optimizer** when historical task data is available. This perspective directly supports our **"task-free" contribution**, enabling optimization strategy learning without requiring pre-trained task distributions.
>
> **3. Structural Distinction from Traditional BBO Methods**
>
> **Regarding the relationship between ABOM and traditional BBO methods**, we would like to highlight a **key structural distinction**. In ABOM, the outer loop represents a **strategy learning problem** ($\min_{\boldsymbol{\theta}} \mathcal{L}^{(t)} = \| \hat{\mathbf{P}}^{(t)} - \mathbf{E}^{(t)} \|^2$), **not a single-task BBO problem**. This self-supervised learning problem is solved using the standard gradient descent method (AdamW). This creates a **fundamental architectural difference** from traditional single-task BBO methods: approaches like DE and NES implement ***single-level optimization*** with **fixed update rules** (e.g., DE's deterministic selection operator and fixed difference operators such as "DE/rand/1", NES's utility function (weighting scheme)-based deterministic selection [1]), whereas ABOM implements ***bilevel optimization*** where the upper level **dynamically adapts update rules** based on optimization progress. **Fig. 4 demonstrates** how ABOM automatically adjusts mutation rules and crossover patterns according to landscape characteristics—a capability **not present in fixed-rule methods**. Furthermore, it is worth noting that our adaptive parameter learning strategy does not restrict the internal structure of the machine learning model, and extending it to existing neural network-based meta-BBO methods is also a very interesting direction.
>
> **4. Experimental Design and Core Contribution Validation**
>
> **Our experimental design aligns with our paper's core contribution** of validating **task-free MetaBBO's effectiveness**. By comparing with established MetaBBO methods, we demonstrate that ABOM achieves **competitive performance without requiring task distributions**, directly addressing our **central research question**. We believe this comparison **appropriately evaluates our contribution** to advancing MetaBBO methodology. Advancing the field of single-task BBO methods is not the motivation of this paper, but it is a valuable topic.
>
> [1] Discovering evolution strategies via meta-black-box optimization. In ICLR, 2023.
>
> We sincerely appreciate your thoughtful feedback, which has helped us significantly improve the clarity and rigor of our paper.

---

> ### Comment · Reviewer_tkpd · 2025-11-22
>
> I thank the authors for their detailed clarification. I now have a clearer understanding of how ABOM performs meta-learning within a single-task setting. However, I maintain that the problem formulation of ABOM remains fundamentally different from traditional MetaBBO methods, which are designed to generalize to unseen tasks after pretraining on a task distribution. In contrast, the primary objective of ABOM is to optimize a single task (in addition to its self-defined learning objective), aligning it more closely with single-task BBO formulations.
>
> Regarding “bilevel optimization”, many single-task BBO algorithms also involve inner–outer optimization structures. For example:
>
> 1. Gaussian-process-based (high-dimensional) Bayesian optimization (e.g., [1]): GP hyperparameters are commonly updated by maximizing the marginal likelihood of the collected samples.
> 2. Neural-network-based optimization methods (e.g., [2]): network parameters are iteratively adapted by minimizing loss functions constructed from sampled data.
>
> For these reasons, I believe it would substantially strengthen the paper to include comparisons with state-of-the-art single-task BBO algorithms, rather than limiting baselines to EA-based MetaBBO methods. Such comparisons would more accurately contextualize ABOM’s performance within the problem setting it ultimately addresses.
> I appreciate the authors’ thorough responses. I have increased my score to 4, but lowered my confidence to 2.
>
> [1] Eriksson, David, et al. "Scalable global optimization via local Bayesian optimization." Advances in neural information processing systems 32 (2019).
>
> [2] Yun, Taeyoung, et al. "Posterior Inference with Diffusion Models for High-dimensional Black-box Optimization." Forty-second International Conference on Machine Learning.

---

> > ### Author Response · Authors · 2025-11-24
> > **Clarification on ABOM's Generalization and Experimental Design**
> >
> > **1. Added Generalization Analysis for ABOM**
> >
> > Thank you for your valuable feedback regarding ABOM's generalization capability. To address this concern, we added a **preliminary generalization analysis** in Appendix I using the STOP benchmark, which contains 12 problems with 10 training tasks and 1 test task per problem. These problems are categorized by **the similarity between training and test tasks** into high (STOP1–4), mixed (STOP5–8), and low (STOP9–12) groups. We introduced **ABOM-PT**, a pre-trained variant that learns from training tasks.
> >
> > Table 6 presents the experimental results of ABOM and ABOM-PT on the STOP benchmark suite, revealing four key insights: 1) ABOM-PT outperforms ABOM in 9 of 12 problems, confirming the generalization capability of our method; 2) Under high-similarity conditions (STOP1-4), ABOM-PT achieves substantially better performance by effectively leveraging optimization knowledge from training tasks to the test task; 3) ABOM-PT underperforms on some mixed-similarity problems (such as STOP8), revealing limitations in handling complex task relationships; 4) Surprisingly, pre-training on low-similarity tasks (STOP9-12) consistently improves performance on the test task, demonstrating that even dissimilar training tasks contain valuable optimization knowledge that enhances generalization capability.
> >
> > **2. Critical clarification on MetaBBO vs. single-task BBO**:
> >
> > **MetaBBO** defines *how* optimization is enhanced through meta techniques, while **single-task BBO** describes *what* problem is being solved: a single black-box problem. **All MetaBBO baselines (EPOM, RLDEAFL, GLEET, etc.) inherently solve single-task BBO problems**—they simply use meta techniques to do so more efficiently. In addition, **we agree that single-task bilevel optimization algorithms are relevant to meta black-box optimization techniques**, and following your suggestion, **we have added reference [1] to our Related Work section** to acknowledge this connection.
> >
> > **3. Critical clarification on MetaBBO vs. ABOM**:
> >
> > **The key distinction of ABOM and MetaBBO lies only in how optimization training data is acquired**: Traditional MetaBBO methods rely on prior task distributions, whereas **ABOM learns dynamically during the optimization process itself**, extending the MetaBBO framework to scenarios where task distributions are unavailable. **This represents a natural progression within MetaBBO rather than a departure from it**.
> >
> > **4. Why our experimental comparison is methodologically sound**:
> >
> > Our evaluation focuses on **comparing ABOM against state-of-the-art MetaBBO methods** because they share the same fundamental objective: enhancing single-task black-box optimization through meta techniques. **All compared methods operate within the MetaBBO paradigm**, differing only in their approach to acquiring the training data. **Our experimental design properly evaluates ABOM within its intended context** as a MetaBBO method, comparing it against the most relevant baselines that solve the same problem with similar objectives.
> >
> > In summary, **ABOM extends the MetaBBO framework by removing the requirement for pre-specified task distributions while maintaining the performance benefits of meta-learning**. The preliminary generalization analysis demonstrates that our approach has meaningful generalization capabilities across tasks, though this was not the primary motivation for our method. **Our experimental comparison appropriately evaluates ABOM against advanced and relevant MetaBBO baselines**, all of which solve single-task BBO problems through meta techniques. We appreciate the opportunity to clarify ABOM's position within the optimization landscape and its relationship to existing MetaBBO approaches.
> >
> > [1] Posterior inference with diffusion models for high-dimensional black-box optimization. In ICML, 2025.

---

### Official Review · Reviewer_kZrh · 2025-10-31

**Soundness:** 3
**Presentation:** 3
**Contribution:** 3
**Rating:** 6
**Confidence:** 4

**Summary:**

This paper proposes an end-to-end optimization model, ABOM, which employs attention mechanisms to perfrom parent selection, crossover and mutation operators on the population to generate offspring. A loss function based on the distance between offspring and the elite archive is introduced to guide the model to improve the offspring. Experimental results validate ABOM's advantage over traditional BBO and MetaBBO methods.

**Strengths:**

1. Unlike existing MetaBBO methods based on deep neural networks, ABOM does not require pre-training, saving substantial time and computational resources. Additionally, the model takes the population itself as input and produces offspring in an end-to-end manner, reducing the human effort required to design optimization states and model actions. Although this end-to-end approach is also adopted in other MetaBBO methods such as RNN-OI and GLHF, ABOM's loss function avoids requiring the gradient of the target problem, which appears novel.

2. The end-to-end neural computation enables efficient batched execution on GPUs, as validated in experiments, but it also imposes higher hardware requirements, especially for high-dimensional problems.

3. ABOM demonstrates superior performance compared to BBO and MetaBBO baselines on both BBOB and UAV path planning tasks, confirming its effectiveness.

**Weaknesses:**

1. The loss function is calculated as the distance between offspring and the elite archive. Given that the search range and problem dimensionality can be large (e.g., [-100, 100] range and 500 dimensions for BBOB in this paper), the scale of the loss and gradients could be large, potentially causing unstable training. Furthermore, the loss function encourages the model to generate offspring that completely surpass the parent population, representing a greedy strategy that may reduce exploration.

2. For each problem instance, model parameters are initialized and updated from scratch. While this enables task-free adaptation, it disregards experience and knowledge gained from optimizing previous problems. Leveraging learned knowledge to enhance current optimization is a key advantage of MetaBBO.

**Questions:**

1. In Figure 4, the selection often focuses on very few individuals (sometimes only one). Does this imply the model is performing a greedy local search around a limited set of points?

2. Are the population sizes and hidden dimensions sensitive to the problem dimension? Figure 5 (a) & (b) indicate that different functions have different optimal settings (e.g., f4 prefers a larger population while f24 prefers a smaller one). Is it feasible to adapt population sizes and hidden dimensions dynamically to improve performance?

---

> ### Author Response · Authors · 2025-11-21
> **Response to Weakness 1–2 and Question 1**
>
> We thank the reviewer for the insightful comments. We have carefully revised the manuscript to address these concerns, with all modifications highlighted in red in the revised version.
>
> **1. Response to Weakness 1 (Training Instability and Greedy Behavior Concerns)**
> We have strengthened our analysis to address both issues:
> - Our revised convergence analysis (**Appendix E**) demonstrates **stable parameter adaptation in high dimensions**. **Fig. 6** shows **consistent loss convergence** across all BBOB test functions ($[-100, 100]^{500}$), with minimal variance across 30 independent runs. This empirical evidence validates our theoretical assumption of local convergence for parameter adaptation (Eq. (18)), confirming that the self-supervised learning paradigm with AdamW optimizer remains stable even in complex optimization scenarios.
>
> - We clarify that our loss function $\min_\theta \|\hat{\mathbf{P}}^{(t)} - \mathbf{E}^{(t)}\|^2$ is **not a greedy strategy but a meta-level learning mechanism**. The "align offspring to elite archive" loss serves to learn better optimization strategies (update rules) at the meta level, not to directly manipulate solutions. Unlike fixed greedy strategies that consistently select the best individuals, ABOM learns **dynamic update rules through attention-based operators**. As shown in **Fig. 4**, our parameterized operators adapt selection and mutation criteria across evolutionary stages.
>
> - The **exploration capability of ABOM** is guaranteed by the attention-based MLP parameterization with dropout, while global convergence is ensured by the combination of exploration capability and elitism (Theorem D.1). Parameter adaptation at the meta level enhances optimization efficiency but does not determine global convergence properties. **Fig. 7** shows that both ABOM and ABOM-NPA (without parameter adaptation) converge, though ABOM achieves better performance. This confirms that **convergence stems from our core mechanisms—exploration capability and elitism—rather than from the meta-level loss function itself**.
>
> **2. Response to Weakness 2 ( Prior Knowledge Integration)**
> We fully agree that leveraging prior task knowledge is a key advantage of MetaBBO:
>
> - We have explicitly enhanced our **future work section** to emphasize this direction: *"Exploring hybrid training paradigms that integrate pretraining on prior knowledge with online adaptation, thereby enhancing optimization efficiency and bridging the gap between task-agnostic adaptation and cross-task generalization."*
>
> - It is worth noting that our **task-free adaptation mechanism and prior knowledge integration are complementary rather than conflicting**. ABOM can incorporate pre-trained knowledge from previous optimization tasks while maintaining its adaptive capabilities. This represents not a limitation but a meaningful extension opportunity: a hybrid training paradigm would allow ABOM to function both as a **task-free optimizer for unknown distributions** and as a **knowledge-enhanced optimizer** when historical task data is available.
>
> **3. Response to Question 1 (Fig. 4 Interpretation Question)**
> We appreciate this insightful question about Fig. 4. The selection patterns do not indicate greedy local search but rather reflect ABOM's dynamically learned optimization strategy:
>
> - **ABOM's optimization strategy** (selection and mutation matrices) **dynamically evolves throughout the optimization process**, while greedy strategies employ fixed rules that never change. In **Fig. 4**, we observe a clear progression: early generations show diverse selection (exploration phase), middle generations focus on promising regions (balancing phase), and later generations concentrate on high-quality solutions (exploitation phase). **This evolution reflects adaptive learning from population information**, unlike fixed greedy methods, which would immediately and permanently focus on the current best solution without any adaptation.
>
> - **ABOM maintains a persistent exploration capability** through our attention-based MLP parameterization with dropout, as theoretically guaranteed by Lemma 1. Even when selection appears concentrated in later stages, our method continues exploring the search space through structured variation mechanisms. In contrast, greedy approaches lack such exploration mechanisms and inevitably get trapped in local optima.

---

> ### Author Response · Authors · 2025-11-21
> **Response to Question 2**
>
> **4. Response to Question 2 (Parameter Sensitivity Question)**
> We thank the reviewer for this important question about parameter sensitivity:
>
> - Our experiments in **Fig. 5** confirm that population sizes and hidden dimensions exhibit sensitivity to different problem characteristics, as different functions indeed show varying optimal settings. However, **ABOM demonstrates strong parameter robustness in practice**. All our experiments across both BBOB and UAV benchmarks use identical parameter settings (Table 5) while achieving state-of-the-art results. **This consistent performance across diverse problems demonstrates ABOM's ability to maintain effectiveness despite parameter sensitivity**.
>
> - We agree that dynamically adapting population size and model capacity represents a valuable research direction. We have added this to **future work** as a promising extension that could further enhance optimization performance.
>
> We sincerely appreciate your thoughtful feedback, which has helped us significantly improve the clarity and rigor of our paper.

---

> > ### Comment · Reviewer_kZrh · 2025-11-22
> >
> > Thanks the authors for the detailed response, which addresses my concerns, so I raise my score to 8. I will be looking forward to your future works.

---

> > > ### Author Response · Authors · 2025-11-24
> > >
> > > We sincerely thank the reviewer for the thoughtful follow-up and for raising the score to 8. We truly appreciate the constructive feedback and the time spent evaluating our work. We are also grateful for your kind anticipation of our future research and look forward to contributing further to the field.

---

### Official Review · Reviewer_RRGJ · 2025-11-01

**Soundness:** 3
**Presentation:** 3
**Contribution:** 3
**Rating:** 6
**Confidence:** 5

**Summary:**

This paper introduces ABOM, By eliminating the dependency on handcrafted training tasks, ABOM performs adaptive parameter learning using only optimization data from the target task itself. This enables zero-shot optimization, where ABOM adapts evolutionary operators during optimization to improve performance dynamically. The paper evaluates ABOM on both synthetic benchmarks (BBOB) and a real-world unmanned aerial vehicle (UAV) path planning problem, demonstrating competitive performance across multiple scenarios. The results are promising, showing that ABOM outperforms existing baseline methods.

**Strengths:**

1. Sound Method Design: The parameterization of evolutionary operators (selection, crossover, mutation) into differentiable, attention-driven modules is well-designed. The online update of optimizer parameters is achieved by using population data generated during the target problem's runtime, with the objective of minimizing the distance between offspring and the elite archive  as the loss function.
2. Theoretical Guarantee (to a degree): The paper provides a convergence proof under idealized assumptions.
3. The experimental evaluation is thorough, and the results are positive.

**Weaknesses:**

1. The paper needs to explain the differences between the proposed method and other, such as GLHF and B2OPT, in detail, especially regarding the parameterization of evolutionary operators. If prior methods were adopted, proper citations are required.
2. The loss function $\min _\theta||\hat{P}^{(t)}-E^{(t)}||_2$ encourages the offspring population $\hat{P}^{(t)}$ to be close to the current elite archive $E^{(t)}$, essentially using the current local optima to guide optimizer updates. This is a form of "bootstrapping," which introduces search bias and may theoretically lead to getting trapped in local optima.
3. While the authors attempted a theoretical proof of global convergence, it is presented in an overly absolute manner. The resolution of the aforementioned bias relies on persistent exploration via Dropout. However, this theory has a problem: its global convergence is inherently "unbounded," meaning there's no guarantee of achieving convergence within a bounded number of iterations. To illustrate, even a completely random algorithm, when combined with a selection operator, can be proven to converge globally. Therefore, the significance of this convergence proof is limited.
4. Furthermore, if the global optimum lies on the boundary of the search space, or if constraint handling cuts the feasible region into discontinuous parts, the convergence proof will not hold. In summary, attempting to prove convergence is commendable, but the conclusions should not be presented too absolutely or rashly. Further refinement and discussion are needed.
5. This paper effectively performs On-Instance Training. Although there's a meta-behavior of "learning optimization strategies," it lacks cross-task generalization, which deviates from the traditional definition of meta-learning. This needs clarification in the introduction.
6. The limitations of On-Instance Training need to be discussed. For example, a lack of adaptation to cold-start scenarios with limited evaluation counts, and the inability to transfer knowledge between tasks (each task requires independent optimization, unable to leverage similar problems for acceleration).
7. A very detailed parameter sensitivity analysis is needed for On-Instance Training. It's recommended to include the learning rate and dropout rate, as they determine the exploration & exploitation tradeoff and are critical to the proposed method.
8. The experimental section lacks in-depth analysis of results. While the performance of ABOM is compared to several baselines, there is limited interpretation of the underlying reasons for ABOM's success.

**Questions:**

Refer to the above weakness part.

---

> ### Author Response · Authors · 2025-11-21
> **Response to Questions 1–4**
>
> We thank the reviewer for your insightful comments. We appreciate your careful review and valuable suggestions for improvement. Below, we address each of your concerns in detail with reference to the specific revisions made in our manuscript. All modifications in the revised manuscript are highlighted in red.
>
> ## Response to Question 1 (Differences with GLHF and B2Opt)
>
> ABOM fundamentally differs from GLHF and B2Opt in three aspects:
> 1) **Structural design**: While GLHF parameterizes Differential Evolution operators (LMM/LCM/SM) and B2Opt uses Transformer-inspired components (SAC) to only model crossover, ABOM implements attention-based operators for *all* evolutionary operations;
> 2) **Learning paradigm**: GLHF and B2Opt require pre-training on task distributions, whereas ABOM performs *task-free adaptation* using only the target task's optimization data;
> 3) **Theoretical foundation**: ABOM provides a rigorous convergence proof (Theorem 1) showing that parameter adaptation does not compromise convergence guarantees—unlike GLHF and B2Opt, which lack such analysis despite empirical success.
>
> ## Response to Question 2 (Explanation of Loss Function)
>
> Thank you for your insightful comment on our loss function $\min_\theta \|\hat{\mathbf{P}}^{(t)} - \mathbf{E}^{(t)}\|^2$. We agree this creates a form of "bootstrapping," but it does not compromise global convergence for the following reasons.
>
> In our revision, we have clarified the strict **bi-level architecture**: loss function (upper level) only guides **how we learn better update rules**, while evolutionary search (lower level) **performs the actual optimization**. As stated in our newly added theoretical contribution section (Appendix D), global convergence depends on lower-level mechanisms: Lemma 1 and Lemma 2 prove persistent exploration via attention-based operators with dropout, and elite preservation ensures the supermartingale property $\mathbb{E}[f_{t+1}^* \mid \mathcal{F}_t] \leq f_t^*$. **This architectural separation ensures that any apparent "bias" in parameter adaptation does not translate into search bias in the actual optimization process.**
>
> To directly address your concern, we have added Fig. 7, showing that both ABOM and ABOM-NPA converge on the BBOB suite. This empirical result confirms that **convergence originates from core mechanisms—elite preservation and attention-based operators with dropout—while parameter adaptation enhances search efficiency**. Furthermore, Theorem D1 rigorously proves that global convergence is maintained **despite parameter adaptation**, requiring only the mild condition (Eq 18): "By continuity of the MLP, there exists $\epsilon > 0$ such that for all $\theta_c \in \mathcal{N}_\epsilon(\theta_c^*)$." This is a standard property of AdamW, empirically validated in Fig. 6.
>
> ## Response to Question 3 (Global Convergence Proof Limitations)
>
>
> We appreciate your observation about the "unbounded" nature of convergence proofs. In response, we have revised our presentation to be more precise:
>
> 1) We have added a discussion on convergence rate analysis to Section 5 (Future Work), acknowledging that **our framework establishes global convergence but does not provide convergence rates.**
>
> 2) In Appendix D, we now state: *"Unlike existing metaBBO methods, ABOM establishes two theoretical foundations: Corollary 1 proves persistent exploration via attention-based MLP parameterization with dropout; Theorem 1 proves global convergence is maintained even with parameter adaptation."*
>
> 3) We explicitly acknowledge that while random search with selection can be proven to converge globally, our key contribution lies in demonstrating that **both persistent exploration and global convergence are preserved despite the inclusion of parameter adaptation mechanisms.**
>
> ## Response to Question 4 (Boundary Conditions and Convergence)
>
> We appreciate your observation regarding boundary conditions. In the revised manuscript, we have:
>
> 1) Explicitly stated in Assumption 1: "The objective $f_T$ is continuous with global minimizer $\mathbf{x}^*$ in the interior of $\mathcal{X}$." This is a **standard assumption in drift analysis for evolutionary algorithms**, consistent with [1].
>
> 2) Added a discussion acknowledging that when the global optimum lies on the boundary or constraint handling induces discontinuous feasible regions, **our current convergence proof does not directly apply.**
>
> [1] *Evolutionary learning: Advances in theories and algorithms.* Springer, 2019.

---

> ### Author Response · Authors · 2025-11-21
> **Response to Questions 5–8**
>
> ## Response to Questions 5 & 6 (Meta-Learning Definition and Task-Free Adaptation)
>
> Thank you for raising these important points about meta-learning definition and adaptation limitations. In the revised manuscript, we have:
>
> 1) Clarified that ABOM qualifies as meta-learning because it learns optimization strategies *during* the optimization process—this is known as *on-instance meta-learning*. The learned parameters $\theta$ represent **a meta-strategy for solving the current task.** Unlike traditional meta-learning, which requires access to a distribution of tasks, ABOM performs *task-free adaptation* while preserving the core principle of meta-learning: **optimizing the optimizer itself.**
>
> 2) Updated our future work statement to comprehensively address these aspects: *"Exploring hybrid training paradigms that integrate **pretraining on prior knowledge** with online adaptation, thereby enhancing optimization efficiency and bridging the gap between task-agnostic adaptation and cross-task generalization."*
>
> This revision clarifies that while our current implementation focuses on task-free adaptation, the framework inherently performs meta-learning by dynamically adapting the optimization strategy based on observed performance. The current limitation of treating each task independently is **not a fundamental constraint**, but rather **an opportunity for meaningful extension through the hybrid training paradigms** outlined above.
>
> ## Response to Question 7 (Parameter Sensitivity Analysis)
>
> We have addressed your concern by adding a comprehensive parameter sensitivity analysis.
>
> For **learning rate sensitivity**, we have added Appendix I with Fig. 9, showing results across the BBOB suite. The analysis reveals that **the adaptation loss converges stably** across learning rates from $1\times10^{-5}$ to $1\times10^{-2}$, while optimization performance varies significantly. A learning rate of $\eta = 1\times10^{-3}$ achieves optimal performance. Although loss convergence is robust, careful selection remains critical for the best overall results.
>
> Regarding **dropout rate analysis**, we have extended Fig. 5 in Section 4.4. Our analysis shows that the crossover ($p_C$) and mutation ($p_M$) dropout parameters achieve optimal performance at $p = 0.95$. **Higher values increase stochasticity and exploration, but setting either to 1 eliminates beneficial randomness, leading to performance degradation.** Controlled stochasticity is essential for maintaining the balance between exploitation and diversity.
>
> These analyses confirm that while parameter adaptation converges reliably, the proper tuning of learning rate and dropout rates is crucial for achieving peak performance.
>
> ## Response to Question 8 (Deeper Analysis of Experimental Results)
>
> We have significantly enhanced our experimental section with the following additions:
>
> 1) **A comprehensive analysis of parameter adaptation** dynamics through Fig. 6 and Fig. 7 (Appendix E), directly addressing your concern about **the limited interpretability of ABOM’s success**. Fig. 6 validates **our theoretical assumption of local convergence in parameter adaptation** by showing stable loss curves. Fig. 7 compares ABOM with ABOM-NPA (without parameter adaptation), demonstrating that while **both methods achieve global convergence, parameter adaptation significantly improves search efficiency.** This empirically confirms that convergence stems from our core mechanisms **—elite preservation and dropout—not solely from adaptation.**
>
> 2) A new comparison with EPOM on the Bipedal Walker task (Appendix H), demonstrating **ABOM’s superior performance over EPOM**, the latest MetaBBO method [1], in the neuroevolution task involving 874 parameters.
>
> 3) The learning rate sensitivity analysis (Appendix I) provides deeper insight into **how parameter adaptation influences optimization performance**, showing that although loss convergence is robust across learning rates, performance is sensitive to their choice.
>
> These additions offer a more thorough understanding of why ABOM succeeds, particularly illustrating **how parameter adaptation enhances search efficiency while preserving essential exploration capabilities.**
>
> We sincerely appreciate your thoughtful feedback, which has helped us significantly improve the clarity and rigor of our paper.
>
> [1] Enhancing zero-shot black-box optimization via pretrained models with efficient population modeling, interaction, and stable gradient approximation. In NeurIPS, 2025.

---

### Official Review · Reviewer_dnfT · 2025-11-01

**Soundness:** 3
**Presentation:** 3
**Contribution:** 3
**Rating:** 8
**Confidence:** 4

**Summary:**

This paper introduces a novel learning-assisted approach for optimization. The authors propose ABOM framework to address the time-consuming training in existing meta-black-box optimization approach. ABOM inherits the concept of  learnable evolutionary operators from previous GLHF framework, where the iterative reproduction & selection operations in an evolutionary algorithm are abstracted as s group of neural networks, and hence bridge efficient gradient decent with learning for optimization. The most important component in ABOM is its online parameter adaption mechanism, where the up-to-date elite solutions are recorded along the optimziation progress, and at each step, the solution population output by the neural network-based evolutionary operators is self-supervised by minimizing the distance with elite solutions at hand. Through this way, ABOM could adapts itself to novel optimization tasks during the online evolution process, without tailored pre-training, The authors provide an intuitive theoretical proof on the convergence of ABOM and validated ABOM's performance on both synthetic and realistic testbeds. The results show that online adaption could deliver comparable performance to existing MetaBBO approaches, while mitigating the training burden.

**Strengths:**

I appreciate the novelty of this paper. Since existing MetaBBO approaches require a pre-defined problem distribution and corresponding pre-training to make the meta-level policy generalizable, the idea in this paper (online adaption through self-supervision) enlights more efficient MetaBBOs.

**Weaknesses:**

1. Theoretical perspective: I have say that though the authors provide a intuitive proof on what they have claimed (Collary 1, 2, Theorem 3.1), a strong assumption (may not happen in real optimization problem) makes me suspect the rationale behind the proof. This assumption is: "the elite solution is ϵ-suboptimal". I wonder how to guarantee such assumption since we face randomized (stocastic) optimization here. This question becomes more obvious when we consider Eq. (10), what if the elite information prematures? What is exact contribution of ABOM to promote the evolution stepping out such premature? Is the crucial component the Dropout in mutation and crossover neural network? Is this indicating that ABOM is actually a random search with adaptive local search?

**Questions:**

See Weaknesses.

---

> ### Author Response · Authors · 2025-11-21
> **Clarification on ABOM's Convergence Analysis and Meta-Learning Framework**
>
> We thank the reviewer for the insightful comments. We have carefully revised the manuscript to address these concerns, with all modifications highlighted in red in the revised version.
>
> **1. Clarification on "ϵ-suboptimal elite" condition**
> This is not an assumption but a conditional statement in our convergence analysis. Lemma 2 and Theorem D.1 rigorously prove global convergence by addressing two complementary cases:
> - When $ f_t^* > f^* + \epsilon $, we establish a positive drift condition $ \mathbb{E}[\Delta V_t \mid \mathcal{F}_t] > \eta(\epsilon) > 0 $.
> - When $ f_t^* \leq f^* + \epsilon $, elitism ensures $ f_{t+1}^* \leq f_t^* $.
>
> This two-case analysis follows the drift analysis methodology in evolutionary optimization theory [1] and does not impose restrictions on the problem structure.
>
> **2. Clarification on Eq. (10) and premature convergence**
> Eq. (10) operates exclusively at the meta-level for **learning optimization strategies, not directly guiding solution generation**:
> - **Meta/base-level separation**: Eq. (10) $ \min_\theta \left\| \hat{P}^{(t)} - E^{(t)} \right\|^2 $ only updates parameters $ \theta $ at the meta-level to learn adaptive optimization strategies (update rules), while the base-level evolutionary process remains fully intact.
> - **Robustness to suboptimal adaptation**: Even if parameter adaptation produces non-optimal $ \theta $, the base-level process maintains exploration through elitism and high-rate dropout ($ p = 0.95 $).
> - Fig. 4 shows that ABOM's selection matrix evolves from diverse exploration to focused exploitation across generations, while the mutation matrix transitions from random initialization to an ordered structure. **This progression confirms Eq. (10) effectively guides the learning of appropriate strategies at different optimization stages.**
> - Fig. 6 shows **consistent parameter adaptation convergence** across the BBOB suite ($ d = 500 $).
> - **Dropout prevents premature convergence** by ensuring persistent exploration, as theoretically guaranteed by Lemma 1.
>
> **3. ABOM vs. random search**
> ABOM fundamentally differs from random search:
> - It learns **structured search patterns** using historical data (Fig. 4), employing different update rules at different optimization stages, unlike uniform randomness. Parameter adaptation is critical: removing it degrades performance (Table 2 and Fig. 7).
> - It consistently outperforms random search across all experiments (Section 4).
>
> The revised manuscript strengthens Appendix D with a clearer explanation of the meta/base-level separation and expands Appendix E to include more comprehensive evidence of parameter adaptation convergence.
>
> [1] *Evolutionary learning: Advances in theories and algorithms.* Springer, 2019.

---

> > ### Comment · Reviewer_dnfT · 2025-11-21
> > **comments on the authors' responses**
> >
> > I appreciate the further clarification provided by the authors. Several further questions come out:
> >
> > 1) I agree that when the case is non-e-optimal, there is a positive drift condition, however, what is the extends of such drift, only positive drift value can not really means we could attain optimal in limited steps? If the authors agree with me, include this explicitly in your paper (limitation or somewhere to let the future readers aware of such issue and the overall context).
> >
> > 2) Since ABOM is an online training paradigm that trains one model for one problem, maybe its generalization potential come from the methodology level? I mean, the difference of MetaBBO and ABOM is that MetaBBO focus on the problem distribution level to train a policy for all, while ABOM tailors policy for a given problem instance. From this perspective, I think if we consider there are hundreds of thousands of problems, ABOM may requires far more resources to train separate policies for these instances while MetaBBO could train on a small subset. I think this point also deserves a clarification in your paper.
> >
> > All in all, I still think this work is an important work in the line of pretrained optimization model, since it renews the training paradigm to mitigate dependence on the gradient of unknown objective functions. I keep my original evaluation and score.

---

> > > ### Author Response · Authors · 2025-11-21
> > > **Response to Questions 1-2**
> > >
> > > We sincerely thank the reviewer for the exceptionally insightful comments and for recognizing that *"this work is an important work in the line of pretrained optimization model, since it renews the training paradigm to mitigate dependence on the gradient of unknown objective functions."* We deeply appreciate you maintaining your original evaluation and score.
> > >
> > > **Response to Question 1 (Convergence Rate Analysis)**
> > >
> > > **We fully agree** that our current drift analysis (Theorem 3.1) establishes global convergence but **does not provide convergence rate bounds** for hitting time.
> > >
> > > **Clarification and action**:
> > > This is an intentional design choice in our theoretical framework:
> > > - Our analysis focuses on **Convergence guarantee** (avoiding permanent premature convergence)
> > > - Convergence rates depend on problem-specific landscape assumptions
> > >
> > > **We will add to the manuscript in Appendix D**:
> > >    *"Theoretical Limitation: Theorem D.1 establishes asymptotic convergence but not polynomial-time convergence. Convergence rate analysis (expected hitting time) for specific problems is one of the important research directions for the future."*
> > >
> > > **Response to Question 2 (ABOM vs. MetaBBO Resource Efficiency)**
> > >
> > > ABOM's task-free adaptation and prior knowledge integration are **complementary capabilities**, not competing approaches. Training ABOM using prior task distributions is a promising future research direction.
> > >
> > > ABOM can incorporate pre-trained knowledge from prior optimization tasks while maintaining its adaptive capabilities. This represents a meaningful extension opportunity: a hybrid training paradigm would allow ABOM to function both as a task-free optimizer for unknown distributions and as a knowledge-enhanced optimizer when historical task data is available.
> > >
> > > **We will add to the manuscript in Section 5**:
> > >
> > > **Hybrid Training with Pre-trained Knowledge**: *"Exploring hybrid training paradigms that integrate pretraining on prior knowledge with online adaptation, thereby enhancing optimization efficiency and bridging the gap between task-agnostic adaptation and cross-task generalization. "*
> > >
> > > Thank you again for your constructive feedback, which will strengthen the clarity of our contribution while preserving the core innovation that you recognized as important to the field.

---

> > > > ### Comment · Reviewer_dnfT · 2025-11-21
> > > > **final comments**
> > > >
> > > > I think my concerns are now fully addressed as the authors have provided how they revise their paper accordingly. I think a score of 8 is the most direct evidence from me to say that this is a good work with valuable contribution.
> > > >
> > > > One little suggestion (I can live with or without it), could you broden related works to include the most recent MetaBBO works to keep the overall literature review scientifically comprehensive?

---

> > > > > ### Author Response · Authors · 2025-11-24
> > > > >
> > > > > We sincerely thank you for confirming that our revisions fully address your concerns and for the score of 8. As suggested, we have added the two recent MetaBBO works to the related works section:
> > > > >
> > > > > - [1] *Instance generation for meta-blackbox optimization through latent space reverse engineering*. arXiv:2509.15810, 2025.
> > > > > - [2] *Posterior inference with diffusion models for high-dimensional black-box optimization*. ICML, 2025.
> > > > >
> > > > > We appreciate this valuable suggestion and thank you for your time and constructive feedback.

---

### Meta-Review · Area_Chair_dqzq · 2025-12-29

**Summary:**

The paper presents a solid contribution to the field of learned optimization. Two reviewers strongly supported it, and one rejected it. However, the rejecting reviewer's concerns were partially addressed during the rebuttal. I recommend acceptance.

**Reviewer Concerns:**

no concerns

**Reviewer Scores:**

not relevant

---

### Decision · Program_Chairs · 2026-01-26

Accept (Oral)